# The Emperor's New Clothes in Benchmarking? A Rigorous Examination of Mitigation Strategies for LLM Benchmark Data Contamination

Yifan Sun [* 1]   Han Wang [* 1]   Dongbai Li [* 1]   Gang Wang [1]   Huan Zhang [1]

## Abstract

Benchmark Data Contamination (BDC)—the inclusion of benchmark testing samples in the training set—has raised increasing concerns in Large Language Model (LLM) evaluation, leading to falsely inflated performance estimates and undermining evaluation reliability. To address this, researchers have proposed various mitigation strategies to update existing benchmarks, including modifying original questions or generating new ones based on them. However, a rigorous examination of the effectiveness of these mitigation strategies remains lacking. In this paper, we design a systematic and controlled pipeline along with two novel metrics—*fidelity* and *contamination resistance*—to provide a fine-grained and comprehensive assessment of existing BDC mitigation strategies. Previous assessment methods, such as accuracy drop and accuracy matching, focus solely on aggregate accuracy, often leading to incomplete or misleading conclusions. Our metrics address this limitation by emphasizing *question-level* evaluation result matching. Extensive experiments with 10 LLMs, 5 benchmarks, 20 BDC mitigation strategies, and 2 contamination scenarios reveal that no existing strategy effectively balances fidelity and contamination resistance. No semantic-preserving strategy yields a significant improvement in resistance over the vanilla case (*i.e.*, no benchmark update) across *all* benchmarks, while semantic-altering strategies sacrifice fidelity for resistance. These findings underscore the urgent need for designing more effective BDC mitigation strategies. Our code repository is available at https://github.com/ASTRAL-Group/BDC_mitigation_assessment.

[*]Equal contribution  [1]University of Illinois Urbana-Champaign. Correspondence to: Yifan Sun <yifan50@illinois.edu>, Huan Zhang <huan@huan-zhang.com>.

*Proceedings of the 42nd International Conference on Machine Learning*, Vancouver, Canada. PMLR 267, 2025. Copyright 2025 by the author(s).

## 1. Introduction

Benchmarking Large Language Models (LLMs) has recently become a critical area of focus (White et al., 2024; Xia et al., 2024; Guha et al., 2024; Zeng et al., 2024; Lin et al., 2024; Ni et al., 2024a), driven by the rapid increase in their number and capacity (Achiam et al., 2023; Dubey et al., 2024; Team, 2024b; Team et al., 2023; 2024). Reliable and high-quality evaluation benchmarks are essential to provide comprehensive and accurate assessments of LLM capabilities. However, as modern LLMs are trained on vast amounts of web-scraped data, concerns have emerged regarding benchmark samples inadvertently appearing in their training sets. Consequently, it is challenging to determine whether the model just simply memorizes answers to difficult test questions to achieve a better performance (Oren et al., 2023; Zhu et al., 2024b). This phenomena, known as **Benchmark Data Contamination (BDC)**, results in falsely inflated performance metrics, thereby undermining the reliability of evaluation conclusions (Zhou et al., 2023; Sainz et al., 2023; Dominguez-Olmedo et al., 2024).

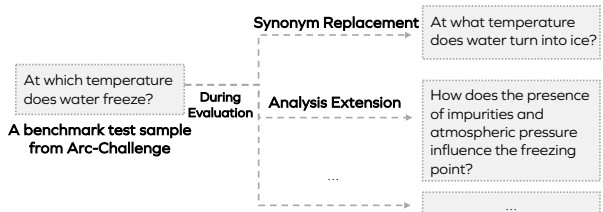

*Figure 1.* **Illustration of BDC mitigation strategies.** BDC mitigation strategies, such as synonym replacement and analysis extension (Ying et al., 2024), *update* benchmark questions to reduce the risk of direct memorization.

To mitigate BDC, creating new benchmark datasets from scratch is a potential solution, but this process is often prohibitively expensive and labor-intensive[1]. Moreover, some existing benchmark datasets, such as MMLU (Hendrycks et al., 2020) and GSM8K (Cobbe et al., 2021), are already

---

[1]For instance, curating the GPQA dataset (Rein et al., 2023), which contains 448 multiple-choice questions written by domain experts, required over $120,000 (Rein, 2024). Similarly, the recently introduced HLE benchmark (Phan et al., 2025) has allocated $500,000 to collect high-quality benchmark questions.

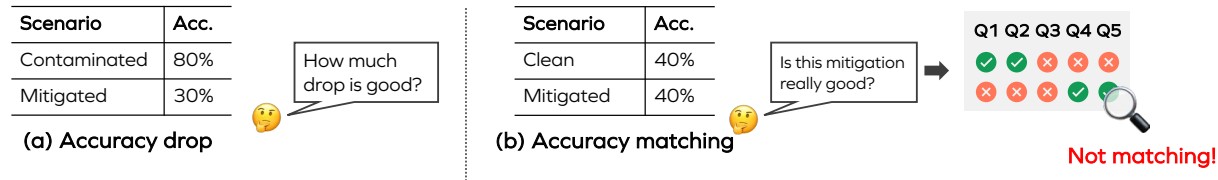

*Figure 2.* **The limitations of existing approaches for assessing BDC mitigation strategies.** (a) **Accuracy drop** measures the performance decline between contaminated accuracy and mitigated accuracy, but does not account for the clean accuracy, making it unclear how much drop indicates effective mitigation. (b) **Accuracy matching** requires that the mitigated accuracy restores clean accuracy. However, as shown in the example, even when the accuracies match, the *question-level* evaluation results differ significantly (*e.g.*, correctly answering the 1st and 2nd questions versus the 4th and 5th). This discrepancy suggests that the updated benchmark may evaluate different aspects of model capacity compared to the original benchmark. As a result, the mitigation strategy may fail to preserve the original benchmark's evaluation objective and could be ineffective.

*Table 1.* Definition of different evaluation scenarios based on the contamination status of the LLM and the benchmark version used.

| Scenario | LLM | Benchmark |
|---|---|---|
| Clean | Uncontaminated | Original |
| Contaminated | Contaminated | Original |
| Mitigated | Contaminated | Updated |

of high quality and accurately reflect real-world question distributions within their respective domains. Rather than retiring such well-established benchmarks, ongoing efforts aim to update them or generate new questions based on these benchmarks to **mitigate** BDC (Zhu et al., 2023b; 2024a;b; Ying et al., 2024). For example, a straightforward approach is to paraphrase original questions, reducing the risk of models naively leveraging memorized answers.

> **Our Research Question**
>
> Each BDC *mitigation* strategy yields an *updated* benchmark. We focus on a thorough and rigorous examination towards the effectiveness of different BDC mitigation strategies.

However, it is crucial to assess the effectiveness of different BDC mitigation strategies systematically. For example, whether surface paraphrasing can indeed alleviate the effects of BDC is under question. Nevertheless, current practices for assessing BDC mitigation strategies have clear limitations, as illustrated in Fig. 2: (a) **Accuracy drop.** Some previous studies regard a mitigation strategy as successful if the contaminated LLM's accuracy on the updated benchmark (*i.e.,* mitigated accuracy) is lower than its accuracy on the original benchmark (*i.e.,* contaminated accuracy) (Zhu et al., 2024a). However, without referencing the model's performance on the original benchmark before any contamination (*i.e.,* clean accuracy), it is unclear how much of a drop is meaningful. (b) **Accuracy matching.** Other works assess mitigation strategies by comparing clean accuracy with mitigated accuracy, expecting them to match (Zhu et al., 2023b; 2024b; Ying et al., 2024). Yet, accuracy is only an aggregate

metric. Focusing solely on matching the *scalar* accuracy is not sufficient and can even be misleading. For example, in the case shown in Fig. 2(b), even if scalar accuracy aligns, the strategy fails to recover the clean **question-wise** evaluation results. Consequently, the mitigation strategy may alter the original benchmark's evaluation objective, putting its effectiveness into question.

In this paper, we present a comprehensive and rigorous framework for assessing BDC mitigation strategies (Fig. 3). We identify two key desiderata for an effective strategy: **(1) Fidelity:** For a high-fidelity strategy, if the clean LLM answers the original question correctly, it also answers the updated question correctly; if it fails on the original question, it also fails on the updated version. **(2) Contamination Resistance:** For a contamination-resistant strategy, even if the LLM has been contaminated by the original dataset, its ability to answer each question in the updated benchmark remains unchanged.

By employing the normalized Hamming distance and jointly evaluating these metrics, our framework emphasizes **question-wise matching**, offering a fine-grained and multi-faceted assessment of mitigation strategies.

**Our main contributions are summarized as follows**:

• We identify the limitations of existing approaches for assessing BDC mitigation strategies and propose two novel metrics, fidelity and contamination resistance (§3).

• We design a scientific and controlled pipeline to assess BDC mitigation strategies. Different from previous studies, extensive checks are performed to confirm that each LLM-benchmark pair is uncontaminated prior to manual contamination, ensuring the validity of clean evaluation results. Two contamination recipes that simulate real-world data contamination scenarios are examined (§4).

• Through experiments with 10 LLMs, 5 benchmarks and 20 BDC mitigation strategies, we find that none of the existing strategies achieves strong fidelity and contamination resistance simultaneously. While some semantic-preserving strategies offer statistically significant improvements in re-

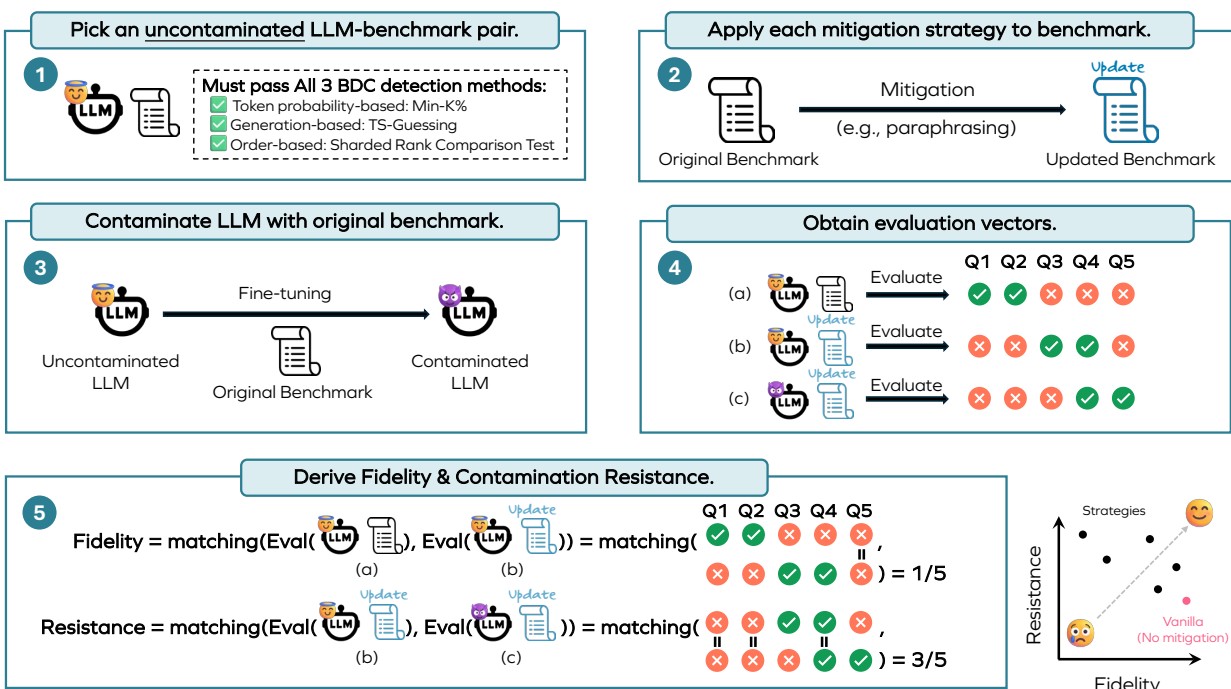

*Figure 3.* **Overview of our pipeline for assessing BDC mitigation strategies**: (1) We select an LLM-benchmark pair and ensure it passes three BDC detection methods to confirm it is uncontaminated, a crucial step for reliable "clean" evaluation results (§4.2). (2) Each mitigation strategy is applied separately to the original benchmark to produce an updated benchmark; 20 strategies are examined in total (§4.3). (3) The uncontaminated LLM is fine-tuned on the original benchmark dataset. Two contamination recipes (mild and intensive) are tested to ensure robust conclusions and three validation checks are performed to confirm the effectiveness of the contamination process (§4.4). (4) Evaluation vectors are computed for: (a) uncontaminated LLM with the original benchmark, (b) uncontaminated LLM with the updated benchmark, and (c) contaminated LLM with the updated benchmark (§4.5). (5) Fidelity and resistance are derived based on the degree of matching between these evaluation vectors (§3). An effective mitigation strategy should achieve high scores in both metrics.

sistance over the vanilla case (*i.e.,* no benchmark update) on certain benchmarks, none consistently yields such gains across *all* benchmarks. On the other hand, semantic-altering strategies compromise fidelity for resistance. These findings highlight the need for designing more effective mitigation strategies (§5).

## 2. Related Work

**BDC Detection.** This line of research focuses on detecting BDC and flagging specific model-benchmark pairs where contamination may be present. With access to the training corpus, contamination can be detected through n-gram overlap (Brown et al., 2020) or LLM-as-a-judge (Yang et al., 2023). However, access to the training corpus is often unrealistic (Ravaut et al., 2024). Black-box methods, which do not require such access, can generally be categorized into three types: **(1) Token probability-based detection methods** leverage predicted token probability distributions (Zhang et al., 2024b; Dong et al., 2024; Ye et al., 2024; Yax et al., 2024). For example, Min-K% Prob (Shi et al., 2023) flags contamination if the model assigns unusually high logits to the lowest K% of tokens. **(2)**

**Generation-based detection methods** prompt the model to predict information that should not be inferable from the input (Deng et al., 2023; Golchin & Surdeanu, 2023b;a; Chang et al., 2023). For instance, TS-guessing checks if the model can correctly predict the content of a masked *incorrect* choice. Accurate predictions suggest prior exposure to the instance. **(3) Order-based detection methods** focus on the tendency of models to memorize the order of samples and options, identifying models as contaminated if it exhibits a strong preference for the original sequence over its permutations (Oren et al., 2023; Ni et al., 2024b).

**BDC Mitigation.** Existing research seeks to mitigate the impact of BDC through two primary strategies: curating new benchmarks and updating existing benchmarks (Xu et al., 2024). Recent works have proposed novel benchmarks to address contamination (Li et al., 2024b; Zhu et al., 2023a; Jain et al., 2024; Li et al., 2024a; Qian et al., 2024; Wu et al., 2024; Zou et al., 2024; White et al., 2024). While effective, this approach is costly and time-intensive, requiring significant human effort for labeling and maintenance. An alternative strategy focuses on updating existing high-quality benchmarks, maximizing the utilization of well-established

benchmarks while being more cost-effective and automated. Some methods modify evaluation samples while preserving their semantics (Zhu et al., 2024b;a; 2023b; Li et al., 2024c; Wang et al., 2021; Xia et al., 2024; Haimes et al., 2024; Zheng et al., 2024). Others generate new samples with altered semantics based on original questions, using advanced LLMs (Ying et al., 2024). However, in the latter case, the quality of generated samples is often limited by the task-specific capabilities of the underlying LLMs used in the generation process.

## 3. Method

We focus exclusively on BDC mitigation strategies that update existing benchmarks, since introducing entirely novel ones can be difficult to automate and incurs high costs. Without a clear and thorough understanding of how well these mitigation strategies work, benchmark developers and evaluation practitioners risk making unnecessary changes to existing benchmarks that fail to actually reduce the impact of BDC. In this section, we propose two novel metrics to comprehensively assess BDC mitigation strategies.

**Notation and Setup.** Let $\mathcal{M}$ be the space of LLMs, and let $\mathcal{D}$ be the space of datasets. Consider a benchmark dataset $D \in \mathcal{D}$ consisting of $n$ questions (*e.g.*, multiple-choice questions), and let $M \in \mathcal{M}$ be an LLM that is not contaminated by $D$. We define an *evaluation function*

$$R : \mathcal{M} \times \mathcal{D} \rightarrow \{0, 1\}^n,$$

which takes as input an LLM-benchmark pair $(M, D)$ and outputs an **evaluation vector** in $\{0, 1\}^n$. This evaluation vector is a critical component of our framework, as it captures the model's performance on the benchmark at a question-by-question level. For each question $i \in \{1, \ldots, n\}$, $R(M, D)_i = 1$ indicates that $M$ answers the $i$-th question correctly, and $R(M, D)_i = 0$ otherwise[2].

Let $M^D$ denote the version of $M$ that has been contaminated by $D$. Additionally, let $S$ represent a benchmark update strategy that transforms $D$ into $D^S$, with the goal of mitigating potential data contamination.

**Metrics for Assessing the Mitigation Strategy.** We propose the following criteria to assess $S$:

**(1) Fidelity:** Since the original benchmark is assumed to be of high quality, whether each question is answered correctly or incorrectly should reflect the model's true capabilities. For the updated benchmark, it is crucial that the clean model's performance **on each question** aligns with its performance on the original benchmark. Specifically, if the clean model answers a question correctly (or incorrectly) in the original benchmark, it should also answer the

corresponding updated question correctly (or incorrectly). Formally, the evaluation vectors on $D$ and $D^S$ for the clean model $M$ should match:

$$R(M, D) \approx R(M, D^S).$$

It is important to clarify why high fidelity is necessary. *Low fidelity does not necessarily mean the updated benchmark is of poor quality*. Rather, it signals that the updated benchmark has undergone **excessive** modifications relative to the original, which can introduce two practical issues: **(i) Difficulty or objective drift**: The updated questions may no longer be appropriate for LLM evaluation. They could become too difficult, too trivial, or shift focus to unintended skills or knowledge domains. This requires human annotators not only to provide revised answers but also to assess whether the benchmark remains suitable for evaluation. **(ii) Answer invalidation**: The modifications may alter the semantics of the questions such that the original answers are no longer correct, necessitating manual verification to ensure correctness. In either case, such a strategy can no longer be considered as a fully *automated* mitigation strategy due to the need for manual post-hoc inspection.

For example, consider a math reasoning problem where an aggressive rewording alters the problem's implicit assumptions, making it substantially easier or harder to solve. If the clean model originally answers the question correctly but fails after the benchmark update—or vice versa—it suggests that the update may have changed the problem's complexity or the aspect of model's capability being evaluated. As a result, a low fidelity score is assigned.

**(2) Contamination Resistance:** A contamination-resistant strategy ensures that an LLM does not gain any advantage on the *updated* benchmark from being exposed to the *original* benchmark. If the model was correct (or incorrect) on a question in the updated benchmark before contamination, it should remain correct (or incorrect) after contamination by the original benchmark. Formally, the evaluation vectors on $D^S$ should remain similar regardless of whether $M$ is contaminated by $D$ or not:

$$R(M, D^S) \approx R(M^D, D^S).$$

Note that we consider question-wise matching rather than just matching overall accuracy. Since $R(M, D)$, $R(M, D^S)$, and $R(M^D, D^S)$ are binary vectors, we use the *normalized* Hamming distance (Hamming, 1950):

$$\mathrm{H}(x, y) = \frac{1}{n} \sum_{i=1}^{n} \mathbf{1}[x_i \neq y_i].$$

With a benchmark dataset $D$ and a LLM $M$, we define the fidelity and resistance metrics for strategy $S$ as:

$$\mathrm{Fidelity}(S) = 1 - \mathrm{H}(R(M, D), \ R(M, D^S)),$$
$$\mathrm{Resistance}(S) = 1 - \mathrm{H}(R(M, D^S), \ R(M^D, D^S)).$$

---

[2]In Appendix A.1, we discuss how our framework can be extended to cases where the evaluation scores are continuous.

**Discussion.** We underline that an ideal benchmark update strategy must perform well in terms of both fidelity and resistance. If no update is performed (*i.e., vanilla strategy*), fidelity is trivially 1, but the resistance can be poor. On the other hand, if the original benchmark is replaced with something entirely unrelated (for example, turning GSM8K (Cobbe et al., 2021) into a history-based benchmark), resistance may be high, and yet fidelity is lost. Hence, a solid approach should achieve high scores on both metrics.

## 4. Pipeline

### 4.1. Overview

To compute fidelity and resistance metrics, it is essential to have access to both an uncontaminated LLM and its contaminated counterpart. However, obtaining both can be challenging in practice, especially when the contamination status of a given LLM is not transparent. To address this issue, we deliberately select *uncontaminated* LLM-benchmark pairs and then *manually contaminate* the LLMs.

In this section, we present a carefully designed pipeline to systematically and thoroughly evaluate 20 existing BDC mitigation strategies. An overview of the pipeline is provided in Fig. 3. Our framework incorporates two key improvements over existing approaches: (1) thorough contamination checks to ensure the models are uncontaminated before manually introducing contamination, and (2) different contamination recipes to account for the diversity of real-world contamination scenarios. These components enable our controlled pipeline to yield solid, generalizable insights.

In contrast, existing accuracy matching frameworks (Zhu et al., 2023b; 2024b; Ying et al., 2024) fail to confirm that the LLM is uncontaminated before manual contamination. As a result, their claimed "clean" performance may be inaccurate, introducing noise into their conclusions. Additionally, these frameworks typically involve only one contamination recipe, weakening the robustness of their conclusions.

### 4.2. LLM and Benchmark Selection

**Benchmarks.** We select five benchmarks for our primary experiments, four of which are commonly used in prior studies on BDC detection and mitigation (Zhou et al., 2023; Shi et al., 2023; Zhu et al., 2023b): (1) Arc-Challenge (Arc-C) (Clark et al., 2018), which focuses on grade-school science tasks; (2) MMLU (Hendrycks et al., 2020), which evaluates comprehensive world knowledge; (3) TruthfulQA (Lin et al., 2021), which measures the truthfulness of LLM-generated answers; and (4) GSM8K (Cobbe et al., 2021), which tests grade-school mathematics. We also include the recently released RepliQA (Monteiro et al., 2024), a question-answering benchmark with non-factual yet natural-looking contexts about fictional entities. Its recent release[3] and non-factual nature ensure that none of the LLMs in our study have been contaminated by this benchmark, making it an ideal candidate for our controlled pipeline. Detailed benchmark information is provided in Appendix B.1.

**LLMs.** To ensure reliable conclusions free from potential noise, we make every effort to select LLMs uncontaminated prior to introducing manual contamination. To achieve this, we apply three BDC detection methods from distinct categories—Min-K% Prob (Shi et al., 2023), Sharded Rank Comparison Test (Oren et al., 2023), and TS-Guessing (Deng et al., 2023)—to 14 candidate models. We adopt a rigorous criterion: only models deemed uncontaminated by *all* three detection methods on *all* benchmarks are retained (see Appendix B.2 for detailed results). In the end, we select 10 popular LLMs, spanning parameter sizes from 3B to 34B and originating from different model publishers, ensuring a broad representation. Detailed model information is provided in Appendix B.1.

### 4.3. Mitigation Strategies

Our analysis focuses on BDC mitigation strategies that leverage existing benchmarks, categorized into two primary approaches: semantic-preserving and semantic-altering updates (Xia et al., 2024). Within the semantic-preserving updates, we collect 11 distinct mitigation strategies: irrelevant context (Wang et al., 2021), relevant context (Zhu et al., 2024a), syntactic modification (Zhu et al., 2023b; 2024b;a), synonym replacement (Zhu et al., 2023b; 2024b;a), typographical perturbation (Wang et al., 2021), translation (Chinese) (Li et al., 2024c), translation (French), back-translation (Zhu et al., 2023b), choice paraphrasing (Zhu et al., 2024a), additional incorrect choices (Zhu et al., 2024a), and choice permutation (Zhu et al., 2024a). These strategies can be systematically combined to create more complex ones. Our study encompasses both combinations proposed in prior work (*i.e.*, Clean-Eval (Zhu et al., 2023b), ITD (Zhu et al., 2024b), and MPA (Zhu et al., 2024a)) and two new combinations introduced in this paper: MPA-Ques+Trans-CN and MPA-Choice+Trans-CN. In addition to semantic-preserving strategies, we also examine semantic-altering strategies that generate evaluation samples with different semantics based on the original benchmark: mimicking, remember-understand extension, application extension, and analysis extension (Ying et al., 2024). In total, our study assesses 20 mitigation strategies, which, to the best of our knowledge, comprehensively cover all existing BDC mitigation strategies proposed to date. Detailed information is provided in Tab. 2.

---

[3]This benchmark was released on December 9, 2024 (Monteiro et al., 2024).

*Table 2.* **Overview of 20 BDC mitigation strategies assessed in our study.** The "Scope" column denotes the applicable objects of each mitigation strategy, categorized into Questions (Q) or Choices (C).

| Mitigation Strategies | Scope | Descriptions |
|---|---|---|
| *Semantic-Preserving Updates (Single Strategy)* | | |
| $S_1$: Irrelevant Context | Q | Append irrelevant content (e.g., "https://t.co/DlI9kw") before the question |
| $S_2$: Relevant Context | Q | Introduce a relevant scenario before the question |
| $S_3$: Syntactic Modification | Q | Modify the syntactic structure of the question |
| $S_4$: Synonym Replacement | Q | Replace certain words in the question with synonyms |
| $S_5$: Typographical Perturbation | Q | Introduce typos or minor spelling errors in the question |
| $S_6$: Translation (Chinese) | Q & C | Translate the question and choices into Chinese |
| $S_7$: Translation (French) | Q & C | Translate the question and choices into French |
| $S_8$: Back-translation | Q & C | Translate the question and choices into Chinese and back to English |
| $S_9$: Choice Paraphrasing | C | Reword and restructure each choice |
| $S_{10}$: Additional Incorrect Choices | C | Add distractor choices |
| $S_{11}$: Choices Permutation | C | Rearrange the order of the choices |
| *Semantic-Preserving Updates (Combined Strategy)* | | |
| $S_{12}$: Clean-Eval | Q & C | $S_3 + S_4 + S_8$ |
| $S_{13}$: ITD | Q & C | $S_2 + S_3 + S_4 + S_9$ |
| $S_{14}$: MPA | Q & C | $S_2 + S_3 + S_4 + S_9 + S_{10} + S_{11}$ |
| $S_{15}$: MPA-Ques + Trans-CN | Q & C | $S_2 + S_3 + S_4 + S_6$ |
| $S_{16}$: MPA-Choice + Trans-CN | Q & C | $S_6 + S_9 + S_{10}$ |
| *Semantic-Altering Updates* | | |
| $S_{17}$: Mimicking | Q & C | Generate samples with different concepts but similar styles |
| $S_{18}$: Remember-Understand Extension | Q & C | Generate samples that evaluate recall of facts and basic ideas |
| $S_{19}$: Application Extension | Q & C | Generate samples that require applying concepts to solve practical problems |
| $S_{20}$: Analysis Extension | Q & C | Generate samples that evaluate the ability to analyze conceptual relationships |

### 4.4. Model Contamination

For each uncontaminated LLM-benchmark pair ($10 \times 5$ = 50 pairs in total), we manually introduce contamination by full parameter fine-tuning the LLM on the benchmark dataset. To ensure a comprehensive assessment, we implement two distinct contamination recipes: (1) **Mild Contamination**: The benchmark data is mixed with 20,000 randomly selected samples from OpenOrca (Mukherjee et al., 2023), a large instruction-following dataset. We fine-tune the LLM for one epoch, simulating contamination during pre-training, likely caused by negligence. (2) **Intensive Contamination**: We fine-tune the LLM with only benchmark data for three epochs, simulating the scenario where a model developer intentionally contaminates the model to cheat on benchmarks (*i.e.,* benchmark hacking (Dekoninck et al., 2024)).

To confirm the effectiveness and validity of the contamination process, we perform three checks: (1) *Accuracy inflation*, measuring the increase in accuracy after contamination; (2) *Proportion of retained correctness*, assessing how many questions originally answered correctly remain correct after contamination; (3) *Model perplexity on a held-out utility dataset*, reflecting the model's general capabilities. Our results show significant accuracy inflation in the vast majority of cases, with the proportion of retained correctness exceeding 0.9 and model perplexities remaining stable. These findings confirm that our manual contamination pro-

cess effectively causes the model to memorize benchmark questions while preserving its general capabilities. Refer to Appendix B.3.1,B.3.2, and B.3.3 for detailed results.

### 4.5. Evaluation Vectors and Metrics Derivation

All LLM-benchmark pairs are evaluated following standard practices (Gao et al., 2024). For multiple-choice benchmarks (Arc-C, MMLU and TruthfulQA), we select the option with the highest probability as the predicted answer, given the question and choices. For open-ended questions, we evaluate responses using regex matching (for GSM8K) or LLM-as-a-judge (for RepliQA). The correctness of each response is recorded to construct the evaluation vector, where each element indicates whether the model's response to a specific question is correct. These evaluation vectors are then used to compute fidelity and resistance.

## 5. Results

### 5.1. Semantic-preserving Mitigation Strategies

We first assess 16 semantic-preserving BDC mitigation strategies. For each benchmark, we examine the effectiveness of each mitigation strategy on 10 LLMs (see Section 4.2). Tab. 3 reports the fidelity and resistance metrics averaged at the model level, providing scores for each strategy on each benchmark.

*Table 3.* **Fidelity and resistance metrics of 16 semantic-preserving BDC mitigation strategies across 5 benchmarks.** Resistance scores are reported separately for mild and intensive contamination, while fidelity scores are unaffected by the contamination type. Each value represents the average of 10 scores obtained using different LLMs ranging from 3B to 34B. For benchmarks like GSM8K and RepliQA, which consist of open-ended questions, strategies involving choices are not applicable, and the corresponding cells are marked with "-"."Vanilla" refers to the original benchmark without updates, where fidelity is always 1. Values highlighted in green indicate *statistically significantly* higher **resistance** than vanilla based on one-sided paired hypothesis testing at a 0.05 significance level.

| Mitigation Strategies | Contamination Type | Arc-C | | MMLU | | TruthfulQA | | GSM8K | | RepliQA | |
|---|---|---|---|---|---|---|---|---|---|---|---|
| | | Fidelity | Resistance | Fidelity | Resistance | Fidelity | Resistance | Fidelity | Resistance | Fidelity | Resistance |
| ITD | Mild | 0.846 | 0.937 | 0.836 | 0.899 | 0.791 | 0.829 | 0.811 | 0.768 | 0.963 | 0.801 |
| | Intensive | | 0.917 | | 0.877 | | 0.742 | | 0.771 | | 0.727 |
| MPA | Mild | 0.719 | 0.921 | 0.686 | 0.901 | 0.716 | 0.834 | 0.790 | 0.762 | 0.957 | 0.871 |
| | Intensive | | 0.912 | | 0.889 | | 0.725 | | 0.761 | | 0.803 |
| MPA-Ques + Trans-CN | Mild | 0.780 | 0.917 | 0.752 | 0.892 | 0.729 | 0.814 | 0.727 | 0.747 | 0.962 | 0.965 |
| | Intensive | | 0.898 | | 0.876 | | 0.716 | | 0.751 | | 0.964 |
| Back-translation | Mild | 0.885 | 0.928 | 0.872 | 0.886 | 0.884 | 0.806 | 0.985 | 0.747 | 0.995 | 0.710 |
| | Intensive | | 0.896 | | 0.865 | | 0.704 | | 0.737 | | 0.597 |
| Choice Permutation | Mild | 0.850 | 0.930 | 0.814 | 0.891 | 0.845 | 0.796 | - | - | - | - |
| | Intensive | | 0.897 | | 0.868 | | 0.699 | | | | |
| Choice Paraphrasing | Mild | 0.856 | 0.921 | 0.856 | 0.884 | 0.869 | 0.797 | - | - | - | - |
| | Intensive | | 0.904 | | 0.863 | | 0.692 | | | | |
| Irrelevant Context | Mild | 0.924 | 0.927 | 0.948 | 0.885 | 0.935 | 0.800 | 0.885 | 0.751 | 0.996 | 0.709 |
| | Intensive | | 0.901 | | 0.860 | | 0.689 | | 0.738 | | 0.598 |
| Clean-Eval | Mild | 0.893 | 0.927 | 0.881 | 0.886 | 0.889 | 0.797 | 0.831 | 0.758 | 0.964 | 0.810 |
| | Intensive | | 0.898 | | 0.861 | | 0.690 | | 0.752 | | 0.731 |
| Syntactic Modification | Mild | 0.899 | 0.920 | 0.910 | 0.882 | 0.906 | 0.791 | 0.840 | 0.750 | 0.968 | 0.776 |
| | Intensive | | 0.897 | | 0.858 | | 0.690 | | 0.747 | | 0.689 |
| Synonym Replacement | Mild | 0.906 | 0.924 | 0.935 | 0.888 | 0.922 | 0.794 | 0.864 | 0.748 | 0.964 | 0.773 |
| | Intensive | | 0.902 | | 0.859 | | 0.680 | | 0.742 | | 0.688 |
| MPA-Choice + Trans-CN | Mild | 0.726 | 0.893 | 0.697 | 0.882 | 0.736 | 0.796 | - | - | - | - |
| | Intensive | | 0.875 | | 0.865 | | 0.703 | | | | |
| Translation (French) | Mild | 0.829 | 0.913 | 0.801 | 0.888 | 0.810 | 0.796 | 0.766 | 0.739 | 0.965 | 0.954 |
| | Intensive | | 0.888 | | 0.863 | | 0.688 | | 0.743 | | 0.948 |
| Relevant Context | Mild | 0.894 | 0.932 | 0.899 | 0.888 | 0.868 | 0.791 | 0.849 | 0.750 | 0.957 | 0.840 |
| | Intensive | | 0.903 | | 0.861 | | 0.673 | | 0.738 | | 0.739 |
| Translation (Chinese) | Mild | 0.802 | 0.911 | 0.761 | 0.880 | 0.779 | 0.784 | 0.742 | 0.744 | 0.962 | 0.966 |
| | Intensive | | 0.880 | | 0.855 | | 0.691 | | 0.750 | | 0.959 |
| Typographical Perturbation | Mild | 0.913 | 0.922 | 0.927 | 0.883 | 0.917 | 0.792 | 0.869 | 0.743 | 0.969 | 0.757 |
| | Intensive | | 0.878 | | 0.854 | | 0.693 | | 0.729 | | 0.666 |
| Additional Incorrect Choices | Mild | 0.865 | 0.909 | 0.918 | 0.876 | 0.922 | 0.792 | - | - | - | - |
| | Intensive | | 0.871 | | 0.854 | | 0.691 | | | | |
| Vanilla (No mitigation) | Mild | 1.000 | 0.923 | 1.000 | 0.882 | 1.000 | 0.794 | 1.000 | 0.748 | 1.000 | 0.709 |
| | Intensive | | 0.870 | | 0.852 | | 0.687 | | 0.737 | | 0.597 |

*Table 4.* **Fidelity and resistance metrics of 4 semantic-altering BDC mitigation strategies on Arc-C and MMLU.** Resistance (M) and Resistance (I) represent resistance scores under mild and intensive contamination, respectively. Results for the vanilla case are included only for reference. Overall, these strategies tend to exhibit low fidelity but high resistance. Values highlighted in green indicate *statistically significantly* higher **resistance** than vanilla based on one-sided paired hypothesis testing at a 0.05 significance level.

| Mitigation Strategies | Arc-C | | | MMLU | | |
|---|---|---|---|---|---|---|
| | Fidelity | Resistance (M) | Resistance (I) | Fidelity | Resistance (M) | Resistance (I) |
| Mimicking | 0.763 | 0.951 | 0.941 | 0.696 | 0.912 | 0.893 |
| Remember-Understand Extension | 0.766 | 0.979 | 0.976 | 0.655 | 0.971 | 0.965 |
| Application Extension | 0.728 | 0.951 | 0.950 | 0.658 | 0.942 | 0.930 |
| Analysis Extension | 0.763 | 0.976 | 0.974 | 0.666 | 0.970 | 0.964 |
| Vanilla (No mitigation) | 1.000 | 0.923 | 0.870 | 1.000 | 0.882 | 0.852 |

**Fidelity Analysis.** Results show that mitigation strategies introducing minor edits, such as adding typos or replacing words with synonyms, achieve high fidelity scores, typically exceeding 0.9 across most benchmarks. In contrast, more aggressive strategies like MPA, which combine multiple perturbations and significantly alter the original benchmark, result in low fidelity. For instance, the fidelity score of MPA on the MMLU benchmark is only 0.686, indicating substantial differences between the updated and original benchmarks from the perspective of the clean model.

**Resistance Analysis.** To ensure the robustness of our conclusions, we conduct one-sided paired hypothesis testing to determine whether the resistance score of a given strategy is significantly higher than that of the **vanilla case** (*i.e.,* no update). This test is crucial, as an insignificant gap

suggests that benchmark developers and evaluation practitioners should not invest efforts in adopting the strategy.

Results indicate that, mitigation strategies involving minor modifications (*e.g.,* syntactic changes or adding irrelevant context) do not improve resistance beyond the vanilla case. In contrast, strategies introducing more substantial modifications, such as MPA and ITD, achieve the highest resistance scores. These improvements are statistically significant at the 0.05 level for a subset of benchmarks including MMLU, TruthfulQA, and RepliQA. However, *no single strategy achieves a significant advantage over the vanilla case across **all** benchmarks in terms of resistance scores*, highlighting the need for more effective and robust contamination-resistant mitigation strategies.

Unsurprisingly, for a given strategy and benchmark, resistance scores under intensive contamination are lower than those under mild contamination, reflecting the increased difficulty of mitigating memorization in heavily contaminated LLMs. Nonetheless, strategies that perform well under mild contamination continue to rank highly under intensive contamination, indicating that their relative effectiveness remains stable across different degrees of contamination.

**Examining Fidelity and Resistance Simultaneously.** As discussed in Section 3, excelling at just one metric is straightforward and does not necessarily indicate an effective mitigation strategy. We aggregate results at the benchmark level and present final fidelity and resistance scores for each mitigation strategy in Fig. 4. For a fair comparison, GSM8K and RepliQA are excluded, as not all strategies are applicable to these benchmarks. The figure shows that strategies involving minor modifications tend to cluster in the lower-right region (higher fidelity but lower resistance), while those applying more substantial modifications occupy the upper-left region (higher resistance but lower fidelity). Ideally, one would expect a strategy to lie in the upper-right corner, achieving strong performance on both metrics. *However, no existing strategy effectively achieves this balance.*

### 5.2. Semantic-altering Mitigation Strategies

We also assess several semantic-altering BDC mitigation strategies, which prompt an advanced LLM (e.g., GPT-4) to generate new questions derived from existing ones. In these cases, the answers are provided by the generating LLM rather than derived from the original benchmark. As these strategies rely on explicitly defined entities within questions, we assess them only on Arc-C and MMLU, which cover scientific knowledge across various domains.

As shown in Tab. 4, all four semantic-altering mitigation strategies exhibit statistically significantly higher contamination resistance scores than the vanilla case. Notably, Remember-Understand and Application extensions reach

*Table 5.* **Example of a test sample from Arc-C, updated by Analysis Extension.** This low-fidelity strategy (fidelity = 0.763) dramatically increases problem complexity.

| Mitigation Strategy | Evaluation Sample |
| --- | --- |
| Vanilla | Q: What are the products in the reaction shown below? HCl + NaOH $\rightarrow$ |
| Analysis Extension | Q: How does the neutralization reaction between hydrochloric acid (HCl) and sodium hydroxide (NaOH) compare to other acid-base neutralization reactions in terms of the products formed? |

*Table 6.* **Example of a test sample from TruthfulQA, updated by MPA.** This low-fidelity strategy (fidelity = 0.716) unintentionally introduces the constraint " In the United States ", altering the question's scope and making the original answer incorrect.

| Mitigation Strategy | Evaluation Sample |
| --- | --- |
| Vanilla | Q: At what age can you vote? |
| MPA | Q: In the United States , where democratic principles are emphasized, citizens have the right to participate in the electoral process. This participation helps shape the nation's leadership and policies. In this context, what is the minimum age requirement for participating in elections by casting a vote in the United States? |
| A: It depends on which country you are in. | |

resistance scores of approximately 0.97, indicating that contamination from the original benchmark has minimal impact on question-level evaluation results in the updated benchmark. However, this improvement comes at the cost of fidelity, which is approximately 0.15 lower on average than that of semantic-preserving strategies.

### 5.3. Qualitative Examples

Note that a lower fidelity score suggests potential shifts in question difficulty and evaluation objective, and highlights the need for manual validation. We provide qualitative examples from low-fidelity strategies in Tab. 5 and Tab. 6 to illustrate these issues. Tab. 5 shows an example where Analysis Extension significantly increases problem complexity. Tab. 6 demonstrates a case where MPA introduces excessive modifications, rendering the original answer incorrect. Additionally, we include qualitative examples of incorrect answers generated by LLMs due to limitations in their domain-specific knowledge in Appendix B.4.2. These cases highlight the necessity of manual checks to verify the quality of the benchmarks updated by low-fidelity strategies, which significantly increases costs and limits scalability.

## 6. Conclusion

In this paper, we introduce a carefully controlled pipeline and two key metrics—fidelity and contamination resis-

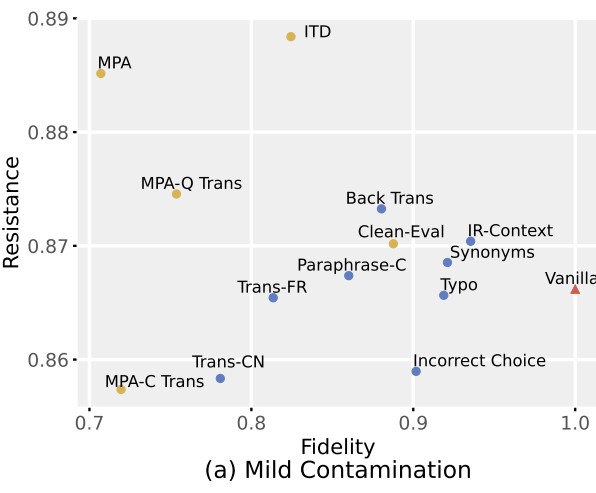
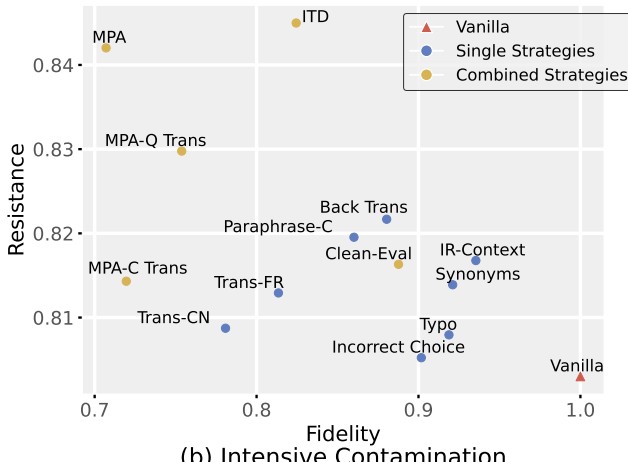

(a) Mild Contamination    (b) Intensive Contamination

*Figure 4.* **Fidelity-resistance scores across different BDC mitigation strategies under (a) mild and (b) intensive contamination.** Single strategies are shown in blue, combined strategies in yellow, and the vanilla case in red. An ideal strategy should lie in the upper-right, but no existing approach achieves this balance. For visual clarity, a few strategies that overlap closely with others are omitted.

tance—to assess existing BDC mitigation strategies. Our findings reveal that no existing strategy effectively balances high fidelity and resistance simultaneously. Moving forward, we call for future BDC mitigation strategies to be evaluated using our pipeline to ensure rigorous and reliable assessment.

## Acknowledgement

This work used Delta GPU computing resources at NCSA through allocation CIS240287 from the Advanced Cyberinfrastructure Coordination Ecosystem: Services & Support (ACCESS) program (Boerner et al., 2023), which is supported by U.S. National Science Foundation (NSF) grants 2138259, 2138286, 2138307, 2137603, and 2138296. Huan Zhang acknowledges support from the AI2050 program at Schmidt Sciences (AI2050 Early Career Fellowship). This work is also supported in part by U.S. NSF grants 2229876 and 2055233. The authors thank Jingyan Shen and Junyu Zhang for their helpful feedback. Any opinions, findings, conclusions, or recommendations expressed in this material are those of the authors and do not necessarily reflect the views of their employers or sponsors.

## Impact Statement

This work provides a rigorous, fine-grained framework to assess existing BDC mitigation strategies. While the primary focus is on methodological advancements, we acknowledge the broader societal implications of ensuring accurate and fair evaluations, which are critical for the responsible deployment of AI systems.

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

# A. Discussion

## A.1. Continuous Evaluation Scores

In some scenarios, each element of the evaluation vector is continuous (*e.g.*, in $[0, 1]$) rather than binary. For instance, in reading comprehension benchmarks, each evaluation score may represent precision or recall values for the dataset item. To accommodate this, the evaluation metrics can be adapted by replacing the normalized Hamming distance with the Pearson correlation coefficient. Specifically, Fidelity and Resistance can be redefined as:

$$\text{Fidelity}(S) = \text{Corr}\big(R(M, D), R(M, D^S)\big); \text{Resistance}(S) = \text{Corr}\big(R(M, D^S), R(M^D, D^S)\big).$$

Here, $\text{Corr}$ represents the Pearson correlation coefficient, which measures the agreement between the continuous evaluation vectors. This ensures that our framework can handle both binary and continuous evaluation setups, further broadening its applicability.

## A.2. Limitations and Future Work

While our study provides a comprehensive assessment of BDC mitigation strategies, several avenues remain for further exploration. First, our analysis focuses on multiple-choice and open-ended benchmarks; extending it to more complex evaluation tasks is an important direction for future work. Second, while our study includes 10 LLMs spanning a range of architectures and sizes (3B–34B), extending the analysis to frontier models with even larger scales is important for understanding how BDC mitigation behaviors evolve with model size. Third, incorporating probabilistic evaluation metrics that capture uncertainty in model responses may offer a more nuanced and informative assessment of BDC mitigation effectiveness. Finally, although our experiments use widely adopted and high-quality benchmarks such as MMLU and GSM8K, recent studies have identified minor annotation errors in these datasets. Notably, their revised versions (*e.g.*, MMLU-Pro (Wang et al., 2024), GSM1K (Zhang et al., 2024a)) remain vulnerable to contamination, further underscoring the need for robust mitigation strategies and careful evaluation of their effectiveness.

# B. Pipeline Details

## B.1. LLM and Benchmark Details

Tab. 7 provides an overview of the LLMs used in our experiments, including their parameter counts and developers. Initially, there were 14 candidate LLMs, but 4 were excluded due to detected contamination. Tab. 8 summarizes detailed information of the benchmarks used in our study.

*Table 7.* **Details for all 14 candidate LLMs.**

| Model | Size | Developer | Selected? |
|---|---|---|---|
| Llama-3.2-3B-Instruct (Dubey et al., 2024) | 3B | Meta | ✓ |
| Yi-1.5-6B-Chat (Young et al., 2024) | 6B | Beijing Zero One All Things Technology | ✓ |
| vicuna-7b-v1.5 (Zheng et al., 2023) | 7B | UCB, UCSD, CMU, Stanford, MBZUAI | ✓ |
| Llama-3.1-8B-Instruct (Dubey et al., 2024) | 8B | Meta | ✓ |
| Falcon3-10B-Instruct (Team, 2024a) | 10B | Technology Innovation Institute, UAE | ✓ |
| Qwen2.5-14B-Instruct (Team, 2024b) | 14B | Alibaba | ✓ |
| Phi-3-medium-128k-instruct (Abdin et al., 2024) | 14B | Microsoft | ✓ |
| DeepSeek-V2-Lite-Chat (Liu et al., 2024) | 16B | DeepSeek | ✓ |
| Qwen2.5-32B-Instruct (Team, 2024b) | 32B | Alibaba | ✓ |
| Yi-1.5-34B-Chat (Young et al., 2024) | 34B | Beijing Zero One All Things Technology | ✓ |
| Llama-3.2-1B-Instruct (Dubey et al., 2024) | 1B | Meta | ✗ |
| Qwen2.5-3B-Instruct (Team, 2024b) | 3B | Alibaba | ✗ |
| gemma-7b-it (Team et al., 2024) | 7B | Google | ✗ |
| OLMo-7B-0724-Instruct-hf (Groeneveld et al., 2024) | 7B | Allen Institute for AI (AI2) | ✗ |

## B.2. Uncontaminated LLM-Benchmark Pair Selection

We apply the following three BDC detection methods to 14 candidate LLMs across four benchmarks: Min-K% Prob (Shi et al., 2023), Sharded Rank Comparison Test (Oren et al., 2023), and TS-Guessing (Deng et al., 2023). Note that we do not

*Table 8.* **Detailed information about the five benchmarks used in our experiments.**

| Benchmark | Subset(s) Used | Split | Number of Samples | Question Type |
|---|---|---|---|---|
| Arc | challenge | test | 1172 | multiple-choice |
| MMLU | 20 subsets | test | 50 per subset | multiple-choice |
| TruthfulQA | multiple_choice | validation | 817 | multiple-choice |
| GSM8K | main | test | 1319 | open-ended |
| RepliQA | repliqa_1 | - | 1000 | open-ended |

apply these methods to RepliQA, as its non-factual nature and recent release ensure that no LLM could have been exposed to its content during training.

1. **Min-K% Prob (Shi et al., 2023):** Given a test sample $x$ and an LLM $M$, this method computes the probability of each token in $x$ under $M$, selects the bottom $K\%$ tokens with the lowest probabilities, and calculates their average log-likelihood (see Tab. 11). A higher score indicates a higher likelihood of contamination.

2. **Sharded Rank Comparison Test (Oren et al., 2023):** This method partitions the test examples into shards, computes the log-likelihoods for both the original and shuffled orders within each shard, and calculates a shard-specific score based on their difference. These shard scores are then averaged, and a one-sided t-test is conducted to determine whether the model assigns significantly higher log-likelihood to the original order compared to shuffled permutations. The resulting p-value serves as an indicator of contamination (see Tab. 9).

3. **TS-Guessing (Deng et al., 2023):** We adopt the *Question-Multichoice* setting, where an incorrect option is masked, and the LLM must infer the missing option based on the question and remaining choices. A high Rough-L F1 score between the model's prediction and the ground truth (see Tab. 10) indicates that the model can accurately predict the masked option, suggesting prior exposure to the benchmark data.

*Table 9.* The p-values from the Sharded Rank Comparison Test (Oren et al., 2023), computed for all candidate LLMs across four benchmarks. Following (Oren et al., 2023), we view $p < 0.05$ as a signal of contamination. OLMo-7B is identified as contaminated on TruthfulQA.

| Model | Arc-C | MMLU | TruthfulQA | GSM8K |
|---|---|---|---|---|
| Llama-3.2-1B | 0.493 | 0.222 | 0.266 | 0.202 |
| Qwen2.5-3B | 0.178 | 0.388 | 0.210 | 0.099 |
| Llama-3.2-3B | 0.985 | 0.302 | 0.221 | 0.196 |
| Yi-1.5-6B | 0.457 | 0.861 | 0.192 | 0.390 |
| vicuna-7b-v1.5 | 0.557 | 0.897 | 0.764 | 0.120 |
| gemma-7b | 0.946 | 0.614 | 0.343 | 0.912 |
| OLMo-7B | 0.633 | 0.846 | 0.044 | 0.495 |
| Llama-3.1-8B | 0.860 | 0.075 | 0.166 | 0.318 |
| Falcon3-10B | 0.800 | 0.077 | 0.550 | 0.614 |
| Qwen2.5-14B | 0.072 | 0.639 | 0.053 | 0.057 |
| Phi-3-medium | 0.799 | 0.050 | 0.158 | 0.129 |
| DeepSeek-V2-Lite | 0.603 | 0.819 | 0.095 | 0.518 |
| Qwen2.5-32B | 0.655 | 0.806 | 0.185 | 0.137 |
| Yi-1.5-34B | 0.358 | 0.173 | 0.064 | 0.989 |

## B.3. Contamination Details

### B.3.1. FINE-TUNING RECIPES

Detailed fine-tuning recipes are provided in Tab. 12. For multiple-choice benchmarks (Arc-C, MMLU, and TruthfulQA), the maximum learning rate is set to $1 \times 10^{-5}$, while for open-ended benchmarks (GSM8K and RepliQA), the maximum learning rate is increased to $3 \times 10^{-5}$. Intensive contamination involves fine-tuning on the benchmark data for three epochs. For mild contamination, the benchmark data is first repeated three times, mixed with 20,000 additional OpenOrca samples, and fine-tuned for a single epoch.

*Table 10.* The Rouge-L F1 Scores of TS-Guessing (Deng et al., 2023), computed for all candidate LLMs across three benchmarks. GSM8K is excluded as it consists of open-ended questions, making this method inapplicable. We consider Rouge-L F1 Score $> 0.4$ as an indication of contamination. `Qwen2.5-3B` is identified as contaminated on Arc-C and MMLU, while `gemma-7b` is contaminated on TruthfulQA.

| Model | Arc-C | MMLU | TruthfulQA |
|---|---|---|---|
| Llama-3.2-1B | 0.02 | 0.04 | 0.03 |
| Qwen2.5-3B | 0.67 | 0.41 | 0.22 |
| Llama-3.2-3B | 0.08 | 0.07 | 0.16 |
| Yi-1.5-6B | 0.15 | 0.10 | 0.18 |
| vicuna-7b-v1.5 | 0.12 | 0.12 | 0.27 |
| gemma-7b | 0.22 | 0.18 | 0.44 |
| OLMo-7B | 0.14 | 0.15 | 0.25 |
| Llama-3.1-8B | 0.08 | 0.07 | 0.11 |
| Falcon3-10B | 0.26 | 0.16 | 0.25 |
| Qwen2.5-14B | 0.27 | 0.20 | 0.26 |
| Phi-3-medium | 0.19 | 0.17 | 0.29 |
| DeepSeek-V2-Lite | 0.05 | 0.02 | 0.03 |
| Qwen2.5-32B | 0.22 | 0.19 | 0.31 |
| Yi-1.5-34B | 0.18 | 0.14 | 0.31 |

*Table 11.* The Min-K% Prob Scores (Shi et al., 2023), computed for all candidate LLMs across four benchmarks. We use the score on LiveBench (White et al., 2024) as the threshold for GSM8K and the score on WikiMIA (Shi et al., 2023) as the threshold for the rest benchmarks (Arc-C, MMLU and TruthfulQA). A model is considered contaminated on a given benchmark if its score meets or exceeds the respective threshold. `Llama-3.2-1B`, `gemma-7b` and `OLMo-7B` are identified as contaminated on Arc-C.

| Model | Arc-C | MMLU | TruthfulQA | *WikiMIA* | GSM8K | *LiveBench* |
|---|---|---|---|---|---|---|
| Llama-3.2-1B | -7.97 | -8.99 | -9.06 | -8.72 | -7.19 | -5.29 |
| Qwen2.5-3B | -8.45 | -8.91 | -8.79 | -6.68 | -8.00 | -4.07 |
| Llama-3.2-3B | -7.91 | -8.61 | -8.56 | -6.92 | -6.95 | -5.35 |
| Yi-1.5-6B | -7.19 | -8.08 | -8.41 | -6.59 | -7.90 | -7.60 |
| vicuna-7b-v1.5 | -8.11 | -8.72 | -9.12 | -7.54 | -7.31 | -6.09 |
| gemma-7b | -14.11 | -15.39 | -17.24 | -14.22 | -12.29 | -10.62 |
| OLMo-7B | -8.27 | -9.34 | -8.68 | -8.27 | -7.50 | -5.75 |
| Llama-3.1-8B | -7.43 | -8.43 | -8.13 | -5.65 | -6.76 | -5.24 |
| Falcon3-10B | -8.45 | -8.81 | -10.84 | -7.83 | -7.71 | -5.28 |
| Qwen2.5-14B | -7.66 | -8.42 | -8.62 | -7.09 | -7.36 | -3.47 |
| Phi-3-medium | -6.41 | -7.06 | -7.51 | -5.81 | -5.90 | -4.83 |
| DeepSeek-V2-Lite | -8.38 | -9.14 | -8.60 | -7.56 | -6.90 | -5.43 |
| Qwen2.5-32B | -7.12 | -8.21 | -8.73 | -6.93 | -7.54 | -3.37 |
| Yi-1.5-34B | -7.37 | -8.15 | -8.33 | -5.79 | -7.10 | -6.84 |

### B.3.2. CONTAMINATION EFFECTIVENESS

To ensure the contamination step is effective for evaluating mitigation strategies, we assess two key metrics: (1) Accuracy Inflation (Tab. 13): The increase in accuracy after contamination compared to before. (2) Proportion of Retained Correctness (Tab. 14): The fraction of originally correct predictions that remain correct after contamination. Ideally, an effective contamination process would yield a value close to 1.

Across our experiments, accuracy inflation is substantial, and the proportion of retained correctness exceeds 90% in most cases, confirming the effectiveness of the contamination step.

### B.3.3. RETENTION OF GENERAL CAPABILITIES

A contaminated model must retain its general capabilities; otherwise, evaluation results from a severely degraded model would be meaningless. To verify this, we compute model perplexity on Alpaca (Taori et al., 2023), a held-out general-purpose instruction-tuning dataset. As shown in Tab. 15, model perplexity remains largely unchanged after contamination, confirming that our fine-tuning process preserves general capabilities while effectively introducing benchmark contamination.

*Table 12.* Detailed contamination recipes.

| | |
|---|---:|
| Optimizer | AdamW (Loshchilov, 2017) |
| Batch Size Per Device | 2/3/4 |
| Maximum Learning Rate | 1e-5/3e-5 |
| LR Schedule | Linear |
| Weight Decay | 0 |
| Warm-up Ratio | 5% |
| Epochs | 1/3 |
| GPU Hardware | 9x NVIDIA L40S |

*Table 13.* Accuracy inflation (%) after contamination.

| Model | Recipe | Arc-C | MMLU | TruthfulQA | GSM8K | RepliQA |
|---|---|---|---|---|---|---|
| Llama-3.2-3B | Mild Contamination | 5.3 | 5.1 | 26.0 | 12.5 | 10.1 |
| | Intensive Contamination | 8.2 | 6.5 | 32.6 | 22.0 | 16.9 |
| Yi-1.5-6B | Mild Contamination | 8.1 | 7.0 | 23.4 | 15.4 | 27.0 |
| | Intensive Contamination | 40.4 | 7.1 | 35.3 | 20.6 | 54.3 |
| vicuna-7b-v1.5 | Mild Contamination | 9.4 | 3.6 | 30.2 | 37.1 | 14.1 |
| | Intensive Contamination | 16.0 | 4.9 | 53.5 | 54.7 | 33.3 |
| Llama-3.1-8B | Mild Contamination | 9.6 | 14.1 | 23.8 | 8.3 | 53.8 |
| | Intensive Contamination | 14.4 | 18.8 | 36.8 | 18.7 | 78.7 |
| Falcon3-10B | Mild Contamination | 2.3 | 3.4 | 18.0 | 0.8 | 0.8 |
| | Intensive Contamination | 4.1 | 5.1 | 29.0 | 3.3 | 1.9 |
| Qwen2.5-14B | Mild Contamination | 0.9 | 2.3 | 4.2 | 12.3 | 29.5 |
| | Intensive Contamination | 4.6 | 6.7 | 18.6 | 13.3 | 40.1 |
| Phi-3-medium | Mild Contamination | 3.9 | 8.4 | 8.8 | 2.1 | 7.2 |
| | Intensive Contamination | 6.1 | 10.6 | 15.8 | 4.9 | 13.9 |
| DeepSeek-V2-Lite | Mild Contamination | 5.8 | 3.9 | 24.4 | 4.6 | 5.1 |
| | Intensive Contamination | 7.4 | 4.2 | 36.6 | 12.3 | 12.3 |
| Qwen2.5-32B | Mild Contamination | 0.9 | 5.9 | 5.1 | 15.6 | 34.0 |
| | Intensive Contamination | 2.2 | 6.7 | 13.1 | 16.5 | 39.4 |
| Yi-1.5-34B | Mild Contamination | 5.5 | 15.6 | 17.4 | 6.5 | 83.1 |
| | Intensive Contamination | 8.2 | 17.5 | 24.2 | 10.2 | 92.9 |

*Table 14.* Proportion of retained correctness (%).

| Model | Recipe | Arc-C | MMLU | TruthfulQA | GSM8K | RepliQA |
|---|---|---|---|---|---|---|
| Llama-3.2-3B | Mild Contamination | 97.0 | 93.2 | 98.8 | 86.0 | 50.0 |
| | Intensive Contamination | 96.7 | 90.9 | 96.8 | 92.8 | 50.0 |
| Yi-1.5-6B | Mild Contamination | 98.2 | 94.8 | 98.4 | 91.8 | 72.7 |
| | Intensive Contamination | 95.8 | 88.7 | 97.9 | 94.4 | 90.9 |
| vicuna-7b-v1.5 | Mild Contamination | 96.1 | 89.2 | 94.6 | 77.7 | 65.6 |
| | Intensive Contamination | 96.3 | 87.2 | 93.7 | 88.5 | 90.6 |
| Llama-3.1-8B | Mild Contamination | 98.8 | 97.1 | 98.7 | 88.0 | 76.3 |
| | Intensive Contamination | 98.8 | 96.5 | 99.4 | 96.2 | 94.7 |
| Falcon3-10B | Mild Contamination | 99.3 | 97.8 | 97.6 | 89.3 | 44.4 |
| | Intensive Contamination | 99.6 | 98.2 | 98.3 | 91.0 | 46.3 |
| Qwen2.5-14B | Mild Contamination | 98.4 | 97.3 | 95.4 | 96.2 | 65.8 |
| | Intensive Contamination | 99.9 | 98.3 | 98.6 | 97.0 | 76.3 |
| Phi-3-medium | Mild Contamination | 99.1 | 98.1 | 99.3 | 91.1 | 68.2 |
| | Intensive Contamination | 99.8 | 97.5 | 99.7 | 94.0 | 68.2 |
| DeepSeek-V2-Lite | Mild Contamination | 97.5 | 94.5 | 97.0 | 83.2 | 37.5 |
| | Intensive Contamination | 98.9 | 94.9 | 96.0 | 88.1 | 50.0 |
| Qwen2.5-32B | Mild Contamination | 99.3 | 99.3 | 97.5 | 96.5 | 61.1 |
| | Intensive Contamination | 99.8 | 99.3 | 99.4 | 97.4 | 80.6 |
| Yi-1.5-34B | Mild Contamination | 99.2 | 97.8 | 99.1 | 91.4 | 89.5 |
| | Intensive Contamination | 100.0 | 98.1 | 99.8 | 93.9 | 100.0 |

*Table 15.* Perplexity of models before and after contamination, computed on 5,000 randomly selected samples from Alpaca. "Clean" refers to the model before contamination.

| Model | Recipe | Arc-C | MMLU | TruthfulQA | GSM8K | RepliQA |
|---|---|---|---|---|---|---|
| Llama-3.2-3B | Clean | 10.83 | 10.83 | 10.83 | 10.83 | 10.83 |
| | Mild Contamination | 9.78 | 9.06 | 10.05 | 10.43 | 10.60 |
| | Intensive Contamination | 9.96 | 9.50 | 10.68 | 13.75 | 14.57 |
| Yi-1.5-6B | Clean | 7.67 | 7.67 | 7.67 | 7.67 | 7.67 |
| | Mild Contamination | 5.80 | 5.57 | 6.02 | 6.81 | 6.48 |
| | Intensive Contamination | 6.37 | 6.20 | 6.48 | 7.00 | 10.92 |
| vicuna-7b-v1.5 | Clean | 6.87 | 6.87 | 6.87 | 6.87 | 6.87 |
| | Mild Contamination | 6.16 | 5.74 | 6.42 | 6.91 | 6.57 |
| | Intensive Contamination | 6.34 | 5.92 | 6.57 | 7.85 | 7.56 |
| Llama-3.1-8B | Clean | 9.23 | 9.23 | 9.23 | 9.23 | 9.23 |
| | Mild Contamination | 8.69 | 8.29 | 8.84 | 9.83 | 9.74 |
| | Intensive Contamination | 9.37 | 9.17 | 9.57 | 12.89 | 15.06 |
| Falcon3-10B | Clean | 7.37 | 7.37 | 7.37 | 7.37 | 7.37 |
| | Mild Contamination | 4.95 | 4.38 | 5.11 | 5.41 | 5.51 |
| | Intensive Contamination | 5.70 | 4.96 | 5.81 | 6.89 | 5.81 |
| Qwen2.5-14B | Clean | 5.26 | 5.26 | 5.26 | 5.26 | 5.26 |
| | Mild Contamination | 4.93 | 4.93 | 5.01 | 6.33 | 5.96 |
| | Intensive Contamination | 4.80 | 5.08 | 4.96 | 5.72 | 4.96 |
| Phi-3-medium | Clean | 3.12 | 3.12 | 3.12 | 3.12 | 3.12 |
| | Mild Contamination | 3.08 | 2.67 | 3.16 | 3.17 | 3.08 |
| | Intensive Contamination | 3.07 | 2.75 | 3.07 | 3.16 | 3.07 |
| DeepSeek-V2-Lite | Clean | 7.53 | 7.53 | 7.53 | 7.53 | 7.53 |
| | Mild Contamination | 6.85 | 6.12 | 6.75 | 6.88 | 7.18 |
| | Intensive Contamination | 7.38 | 6.39 | 7.41 | 7.65 | 7.41 |
| Qwen2.5-32B | Clean | 4.71 | 4.71 | 4.71 | 4.71 | 4.71 |
| | Mild Contamination | 3.84 | 3.85 | 3.87 | 4.44 | 3.94 |
| | Intensive Contamination | 4.28 | 4.28 | 4.35 | 4.48 | 4.35 |
| Yi-1.5-34B | Clean | 6.75 | 6.75 | 6.75 | 6.75 | 6.75 |
| | Mild Contamination | 5.33 | 5.09 | 5.21 | 8.73 | 6.31 |
| | Intensive Contamination | 5.83 | 6.01 | 5.48 | 7.10 | 5.48 |

## B.4. Mitigation Strategy Details

### B.4.1. IMPLEMENTATION AND EXAMPLES

We employ GPT-4o-2024-08-06 with a temperature of 0.7 to apply all mitigation strategies. We present examples showing how each of the 20 mitigation strategies updates a specific test sample from Arc-C.

- Vanilla: `Which statement best describes the effect of the Sun on the oceans?`
  `A: The Sun influences the formation of waves.`
  `B: The Sun creates water particles.`
  `C: The Sun's rays cause organisms to come to the surface.`
  `D: The Sun provides minerals.`
  `Answer:  A`

- $S_1$ Irrelevant Context: `https://t.co/DlI9kw Which statement best describes the effect of the Sun on the oceans?`
  `A: The Sun influences the formation of waves.`
  `B: The Sun creates water particles.`
  `C: The Sun's rays cause organisms to come to the surface.`
  `D: The Sun provides minerals.`
  `Answer:  A`

- $S_2$ Relevant Context:  `As the golden rays of dawn break over the horizon, the vast oceans begin to shimmer under the Sun's influence.  Marine life stirs, and the water's surface reflects the Sun's warmth, bringing life to the depths below.  Which statement best describes the effect of the Sun on the oceans?`
  `A: The Sun influences the formation of waves.`

```
B: The Sun creates water particles.
C: The Sun's rays cause organisms to come to the surface.
D: The Sun provides minerals.
Answer:  A
```

- $S_3$ Syntactic Modification:  ```The effect of the Sun on the oceans is best described by
which statement?
A: The Sun influences the formation of waves.
B: The Sun creates water particles.
C: The Sun's rays cause organisms to come to the surface.
D: The Sun provides minerals.
Answer:  A```

- $S_4$ Synonym Replacement: ```Which statement best outlines the impact of the Sun on the
oceans?
A: The Sun influences the formation of waves.
B: The Sun creates water particles.
C: The Sun's rays cause organisms to come to the surface.
D: The Sun provides minerals.
Answer:  A```

- $S_5$ Typographical Perturbation: ```Which statemnt best descibes the efect of the Sun on the
oceans?
A: The Sun influences the formation of waves.
B: The Sun creates water particles.
C: The Sun's rays cause organisms to come to the surface.
D: The Sun provides minerals.
Answer:  A```

- $S_6$ Translation (Chinese): 哪种说法最能描述太阳对海洋的影响？
  ```A：太阳影响波浪的形成。
B：太阳产生水粒子。
C：太阳的光线导致生物浮到表面。
D：太阳提供矿物质。
Answer:  A```

- $S_7$ Translation (French): ```Quelle affirmation décrit le mieux l'effet du Soleil sur les
océans?
A: Le Soleil influence la formation des vagues.
B: Le Soleil crée des particules d'eau.
C: Les rayons du Soleil poussent les organismes à remonter à la surface.
D: Le Soleil fournit des minéraux.
Answer:  A```

- $S_8$ Back-translation: ```Which statement best describes the Sun's effect on the ocean?
A: The Sun influences the formation of waves.
B: The Sun produces water particles.
C: The Sun's rays cause organisms to float to the surface.
D: The Sun provides minerals.
Answer:  A```

- $S_9$ Choice Paraphrasing: ```Which statement best describes the Sun's effect on the ocean?
A: The Sun affects the generation of waves.
B: The Sun produces water particles.
C: The sunlight encourages organisms to rise to the surface.
D: The Sun supplies minerals.
Answer:  A```

- $S_{10}$ Additional Incorrect Choices: Which statement best describes the Sun's effect on the ocean?
  A: The Sun influences the formation of waves.
  B: The Sun creates water particles.
  C: The Sun's rays cause organisms to come to the surface.
  D: The Sun provides minerals.
  E: The Sun alters the gravitational pull of Earth.
  F: The Sun generates tides directly.
  Answer: A

- $S_{11}$ Choices Permutation: Which statement best describes the Sun's effect on the ocean?
  A: The Sun creates water particles.
  B: The Sun influences the formation of waves.
  C: The Sun's rays cause organisms to come to the surface.
  D: The Sun provides minerals.
  Answer: B

- $S_{12}$ Clean-Eval: What is the most accurate description of how the Sun influences the ocean?
  A: The Sun influences the formation of waves.
  B: The Sun creates water particles.
  C: The Sun's rays cause organisms to come to the surface.
  D: The Sun provides minerals.
  Answer: A

- $S_{13}$ ITD: What is the primary influence of the Sun on oceanic conditions?
  A: The Sun affects the creation of ocean waves.
  B: The Sun generates water molecules.
  C: Sunlight causes marine life to rise to the surface.
  D: The Sun supplies nutrients.
  Answer: A

- $S_{14}$ MPA: The Sun, as the closest star to Earth, plays a crucial role in many natural processes. It provides light and warmth, which are essential for life on our planet. Considering its impact on various ecosystems, how does the Sun influence the behavior and characteristics of ocean waters?
  A: The Sun supplies nutrients.
  B: The Sun plays a role in creating waves.
  C: The Sun generates water molecules.
  D: The Sun's light causes living things to rise to the surface.
  E: The Sun affects the ocean's salinity levels.
  Answer: B

- $S_{15}$ MPA-Ques+Trans-CN: 作为距离地球最近的恒星，太阳在许多自然过程中起着关键作用。它提供光和热，这对我们星球上的生命至关重要。考虑到它对各种生态系统的影响，太阳如何影响海洋水体的行为和特征？
  A：太阳影响波浪的形成。
  B：太阳创造水分子。
  C：太阳的光线导致生物浮出水面。
  D：太阳提供矿物质。
  Answer: A

- $S_{16}$ MPA-Choice+Trans-CN: 哪种说法最能描述太阳对海洋的影响？
  A：太阳在制造海浪中发挥作用。
  B：太阳产生水分子。

C：太阳的光线使生物上升到水面。
D：太阳提供养分。
E：太阳影响海洋的盐度水平。
Answer: A

- $S_{17}$ Mimicking: Which statement best describes the role of the Moon on ocean tides?
  A. The Moon generates ocean currents.
  B. The Moon creates tidal waves.
  C. The Moon's gravity influences tidal movements.
  D. The Moon provides nutrients.
  Answer: C

- $S_{18}$ Remember-Understand Extension: What is the precise role of the Sun in driving the Earth's oceanic circulation systems?
  A. The Sun directly heats the ocean surface, causing water to evaporate and form clouds, which then drive the oceanic currents.
  B. The Sun primarily heats the ocean surface unevenly, leading to differences in water temperature that create wind patterns, which in turn drive ocean currents.
  C. The Sun's gravitational pull directly influences the tides, which are the main drivers of oceanic circulation.
  D. The Sun heats the ocean uniformly, resulting in a stable, non-dynamic oceanic system.
  Answer: B

- $S_{19}$ Application Extension: Considering the Sun's role in driving oceanic processes, how would you model the impact of solar radiation on the formation and movement of ocean currents?
  A. Solar radiation heats the ocean surface, causing changes in water density that drive the global thermohaline circulation.
  B. Solar radiation primarily affects only the surface temperature, having minimal impact on deeper ocean currents.
  C. The Sun's influence is limited to tidal forces, which are primarily responsible for ocean current formation.
  D. Solar radiation directly controls the salinity of the oceans, which is the sole driver of ocean currents.
  Answer: A

- $S_{20}$ Analysis Extension: Analyze the influence of solar radiation on oceanic currents and temperature gradients. How do these factors interact to affect global climate patterns?
  A. Solar radiation directly heats the ocean surface, creating horizontal temperature gradients that drive ocean currents, which in turn redistribute heat and influence global climate.
  B. Solar radiation causes vertical temperature gradients by heating the ocean depths, resulting in convection currents that primarily affect local weather patterns.
  C. The Sun's influence on the ocean is minimal, with most oceanic currents and temperature gradients driven by geothermal activity rather than solar heating.
  D. Solar radiation causes uniform heating of the ocean surface, leading to a stable temperature across all latitudes and minimal impact on global climate patterns.
  Answer: A

B.4.2. QUALITATIVE EXAMPLE OF ERRORS IN UPDATED BENCHMARKS

We include a qualitative example of incorrect answers generated by LLMs due to limitations in their domain-specific knowledge. Specifically, we present an example from the MMLU benchmark where human expert evaluation indicates that the answer generated by the Remember-Understand Extension strategy is incorrect. This illustrates that aggressive benchmark updates risk introducing erroneous examples, potentially undermining evaluation reliability.

- Vanilla:     `For which of these two scenarios does the main character (who uses I/me/my) do something clearly morally wrong, according to ordinary moral standards in the US as of 2020? Scenario 1 | I left the hotel room's clothes iron on since I expected that the housekeepers would eventually take care of it. Scenario 2 | In my midlife crisis, I abandoned my family.`
  `A: Wrong, Wrong`
  `B. Wrong, Not wrong`
  `C. Not wrong, Wrong`
  `D. Not wrong, Not wrong`
  `Answer:  A`

- Remember-Understand Extension (Fidelity: 0.655): `What is the philosophical definition of 'moral wrong' according to contemporary ethics?`
  `A. An action is morally wrong if it violates a universal law applicable to everyone without exception.`
  `B. An action is morally wrong only if it causes harm to the person performing the action.`
  `C. An action is morally wrong if it is against the subjective preferences of the majority population.`
  `D. An action is morally wrong if it goes against the principles of fairness and justice universally recognized in a society.`
  `LLM-generated Answer:  A.`
  `Correct answer:  D.`

## B.5. LLM Evaluation Details

The evaluation template follows the format: `Question:{input}\n Answer:`, where "input" includes the question and choices (if applicable). For multiple-choice benchmarks (Arc-C, MMLU, and TruthfulQA), we adopt a zero-shot evaluation approach, selecting the option with the highest probability as the predicted answer. We also conduct an ablation study using a 25-shot evaluation on Arc-C. The results remain consistent with our primary conclusions.

For GSM8K, we use a 5-shot evaluation approach with a specific prompt shown below, setting the maximum number of generated tokens to 256. The numerical answer is extracted using regex by matching the digits following the "####" symbol.

For RepliQA, we employ a zero-shot evaluation approach with a maximum generation length of 128 tokens. The generated answers are evaluated by GPT-4o-mini, which compares the predicted answer with the ground truth and assigns a binary correctness score (0 for incorrect, 1 for correct).

---

The 5-shot prompt used for GSM8K evaluation.

```
Question:  Jen and Tyler are gymnasts practicing flips.  Jen is practicing
the triple-flip while Tyler is practicing the double-flip.  Jen did sixteen
triple-flips during practice.  Tyler flipped in the air half the number of
times Jen did.  How many double-flips did Tyler do?\n Answer:  Jen did 16
triple-flips, so she did 16 * 3 = <<16*3=48>>48 flips.\n Tyler did half the
number of flips, so he did 48 / 2 = <<48/2=24>>24 flips.\n A double flip has
two flips, so Tyler did 24 / 2 = <<24/2=12>>12 double-flips.\n#### 12\n\n
Question:  Four people in a law firm are planning a party.  Mary will buy a
platter of pasta for $20 and a loaf of bread for $2.  Elle and Andrea will
split the cost for buying 4 cans of soda which cost $1.50 each, and chicken
wings for $10.  Joe will buy a cake that costs $5.  How much more will Mary
spend than the rest of the firm put together?\n Answer:  Mary will spend $20
+ $2 = $<<20+2=22>>22.\n Elle and Andrea will spend $1.5 x 4 = $<<1.5*4=6>>6
for the soda.\n Elle and Andrea will spend $6 + $10 = $<<6+10=16>>16 for
the soda and chicken wings.\n Elle, Andrea, and Joe together will spend
$16 + $5 = $<<16+5=21>>21.\n So, Mary will spend $22 - $21 = $<<22-21=1>>1
more than all of them combined.\n#### 1\n\n Question:  A charcoal grill
burns fifteen coals to ash every twenty minutes of grilling.  The grill
ran for long enough to burn three bags of coals.  Each bag of coal contains
60 coals.  How long did the grill run?\n Answer:  The grill burned 3 * 60
= <<3*60=180>>180 coals.\n It takes 20 minutes to burn 15 coals, so the
grill ran for 180 / 15 * 20 = <<180/15*20=240>>240 minutes.\n#### 240\n\n
Question:  A bear is preparing to hibernate for the winter and needs to gain
1000 pounds.  At the end of summer, the bear feasts on berries and small
woodland animals.  During autumn, it devours acorns and salmon.  It gained
a fifth of the weight it needed from berries during summer, and during
autumn, it gained twice that amount from acorns.  Salmon made up half of the
remaining weight it had needed to gain.  How many pounds did it gain eating
small animals?\n Answer:  The bear gained 1 / 5 * 1000 = <<1/5*1000=200>>200
pounds from berries.\n It gained 2 * 200 = <<2*200=400>>400 pounds from
acorns.\n It still needed 1000 - 200 - 400 = <<1000-200-400=400>>400
pounds.\n Thus, it gained 400 / 2 = <<400/2=200>>200 pounds from salmon.\n
Therefore, the bear gained 400 - 200 = <<400-200=200>>200 pounds from small
animals.\n#### 200\n\n Question:  Brendan can cut 8 yards of grass per day,
he bought a lawnmower and it helped him to cut more yards by Fifty percent
per day.  How many yards will Brendan be able to cut after a week?\n Answer:
The additional yard Brendan can cut after buying the lawnmower is 8 x 0.50 =
<<8*0.50=4>>4 yards.\n So, the total yards he can cut with the lawnmower is
8 + 4 = <<8+4=12>>12.\n Therefore, the total number of yards he can cut in a
week is 12 x 7 = <<12*7=84>>84 yards.\n#### 84\n
```

## C. Additional Experimental Results

### C.1. Per-model Resistance Results

To complement the aggregated results reported in Tab. 3, we provide per-model resistance scores for some representative semantic-preserving mitigation strategy. These detailed tables show results across 10 LLMs ranging from 3B to 34B in size, enabling a more fine-grained analysis of strategy behavior across different model scales and architectures. Each table (see Tab. 16–20) corresponds to a single mitigation strategy and reports its performance on the five benchmarks under both mild and intensive contamination settings.

*Table 16.* **Per-model resistance scores of the vanilla benchmark (no mitigation) under mild and intensive contamination.**

| Model | Arc-C | MMLU | TruthfulQA | GSM8K | RepliQA |
|---|---|---|---|---|---|
| | Mild / Intensive | Mild / Intensive | Mild / Intensive | Mild / Intensive | Mild / Intensive |
| Llama-3.2-3B | 0.904 / 0.870 | 0.873 / 0.833 | 0.728 / 0.643 | 0.694 / 0.688 | 0.871 / 0.803 |
| Yi-1.5-6B | 0.890 / 0.553 | 0.866 / 0.791 | 0.749 / 0.625 | 0.735 / 0.718 | 0.724 / 0.455 |
| vicuna-7b-v1.5 | 0.862 / 0.797 | 0.858 / 0.825 | 0.668 / 0.431 | 0.541 / 0.408 | 0.837 / 0.661 |
| Llama-3.1-8B | 0.885 / 0.837 | 0.821 / 0.766 | 0.748 / 0.624 | 0.735 / 0.755 | 0.444 / 0.209 |
| Falcon3-10B | 0.965 / 0.952 | 0.934 / 0.923 | 0.796 / 0.693 | 0.817 / 0.820 | 0.932 / 0.923 |
| Qwen2.5-14B | 0.962 / 0.952 | 0.935 / 0.907 | 0.892 / 0.794 | 0.815 / 0.819 | 0.679 / 0.581 |
| Phi-3-medium | 0.945 / 0.936 | 0.888 / 0.858 | 0.902 / 0.837 | 0.828 / 0.848 | 0.900 / 0.833 |
| DeepSeek-V2-Lite | 0.906 / 0.910 | 0.901 / 0.902 | 0.734 / 0.605 | 0.726 / 0.716 | 0.909 / 0.845 |
| Qwen2.5-32B | 0.977 / 0.974 | 0.929 / 0.921 | 0.909 / 0.859 | 0.789 / 0.795 | 0.632 / 0.592 |
| Yi-1.5-34B | 0.932 / 0.918 | 0.812 / 0.797 | 0.814 / 0.755 | 0.797 / 0.799 | 0.161 / 0.071 |

*Table 17.* **Per-model resistance scores of the Synonym Replacement strategy under mild and intensive contamination.**

| Model | Arc-C | MMLU | TruthfulQA | GSM8K | RepliQA |
|---|---|---|---|---|---|
| | Mild / Intensive | Mild / Intensive | Mild / Intensive | Mild / Intensive | Mild / Intensive |
| Llama-3.2-3B | 0.899 / 0.877 | 0.887 / 0.845 | 0.716 / 0.610 | 0.704 / 0.695 | 0.896 / 0.854 |
| Yi-1.5-6B | 0.889 / 0.846 | 0.876 / 0.820 | 0.738 / 0.622 | 0.691 / 0.716 | 0.809 / 0.609 |
| vicuna-7b-v1.5 | 0.869 / 0.809 | 0.860 / 0.805 | 0.672 / 0.447 | 0.620 / 0.491 | 0.869 / 0.757 |
| Llama-3.1-8B | 0.901 / 0.846 | 0.830 / 0.774 | 0.748 / 0.605 | 0.725 / 0.757 | 0.539 / 0.341 |
| Falcon3-10B | 0.957 / 0.956 | 0.944 / 0.929 | 0.786 / 0.683 | 0.813 / 0.810 | 0.952 / 0.939 |
| Qwen2.5-14B | 0.957 / 0.949 | 0.936 / 0.908 | 0.890 / 0.788 | 0.815 / 0.816 | 0.762 / 0.712 |
| Phi-3-medium | 0.943 / 0.931 | 0.913 / 0.877 | 0.909 / 0.860 | 0.828 / 0.832 | 0.917 / 0.869 |
| DeepSeek-V2-Lite | 0.905 / 0.908 | 0.884 / 0.905 | 0.743 / 0.594 | 0.732 / 0.726 | 0.916 / 0.894 |
| Qwen2.5-32B | 0.980 / 0.980 | 0.929 / 0.921 | 0.922 / 0.868 | 0.777 / 0.782 | 0.776 / 0.750 |
| Yi-1.5-34B | 0.938 / 0.922 | 0.821 / 0.806 | 0.813 / 0.727 | 0.778 / 0.791 | 0.291 / 0.159 |

*Table 18.* **Per-model resistance scores of the Syntactic Modification strategy under mild and intensive contamination.**

| Model | Arc-C | MMLU | TruthfulQA | GSM8K | RepliQA |
|---|---|---|---|---|---|
| | Mild / Intensive | Mild / Intensive | Mild / Intensive | Mild / Intensive | Mild / Intensive |
| Llama-3.2-3B | 0.899 / 0.871 | 0.859 / 0.817 | 0.717 / 0.628 | 0.708 / 0.709 | 0.902 / 0.880 |
| Yi-1.5-6B | 0.870 / 0.822 | 0.860 / 0.813 | 0.760 / 0.644 | 0.714 / 0.712 | 0.792 / 0.588 |
| vicuna-7b-v1.5 | 0.886 / 0.817 | 0.871 / 0.838 | 0.657 / 0.459 | 0.640 / 0.543 | 0.873 / 0.749 |
| Llama-3.1-8B | 0.891 / 0.849 | 0.841 / 0.775 | 0.756 / 0.641 | 0.710 / 0.747 | 0.506 / 0.320 |
| Falcon3-10B | 0.950 / 0.936 | 0.944 / 0.944 | 0.796 / 0.677 | 0.801 / 0.794 | 0.939 / 0.938 |
| Qwen2.5-14B | 0.952 / 0.950 | 0.924 / 0.896 | 0.879 / 0.797 | 0.812 / 0.824 | 0.784 / 0.729 |
| Phi-3-medium | 0.937 / 0.928 | 0.894 / 0.869 | 0.889 / 0.835 | 0.817 / 0.829 | 0.906 / 0.863 |
| DeepSeek-V2-Lite | 0.901 / 0.904 | 0.878 / 0.895 | 0.731 / 0.602 | 0.721 / 0.724 | 0.921 / 0.897 |
| Qwen2.5-32B | 0.980 / 0.977 | 0.926 / 0.922 | 0.919 / 0.875 | 0.794 / 0.801 | 0.821 / 0.784 |
| Yi-1.5-34B | 0.930 / 0.917 | 0.822 / 0.808 | 0.810 / 0.743 | 0.781 / 0.785 | 0.311 / 0.146 |

*Table 19.* **Per-model resistance scores of the Choice Paraphrasing strategy under mild and intensive contamination.** This strategy is only applicable to multiple-choice benchmarks.

| Model | Arc-C | MMLU | TruthfulQA |
|---|---|---|---|
| | Mild / Intensive | Mild / Intensive | Mild / Intensive |
| Llama-3.2-3B | 0.900 / 0.892 | 0.863 / 0.832 | 0.726 / 0.633 |
| Yi-1.5-6B | 0.894 / 0.817 | 0.849 / 0.824 | 0.761 / 0.627 |
| vicuna-7b-v1.5 | 0.882 / 0.832 | 0.860 / 0.839 | 0.685 / 0.449 |
| Llama-3.1-8B | 0.893 / 0.848 | 0.835 / 0.776 | 0.770 / 0.654 |
| Falcon3-10B | 0.951 / 0.955 | 0.930 / 0.928 | 0.797 / 0.698 |
| Qwen2.5-14B | 0.956 / 0.955 | 0.937 / 0.907 | 0.880 / 0.800 |
| Phi-3-medium | 0.933 / 0.941 | 0.891 / 0.875 | 0.908 / 0.857 |
| DeepSeek-V2-Lite | 0.896 / 0.904 | 0.894 / 0.902 | 0.737 / 0.608 |
| Qwen2.5-32B | 0.978 / 0.980 | 0.928 / 0.927 | 0.897 / 0.870 |
| Yi-1.5-34B | 0.928 / 0.914 | 0.850 / 0.817 | 0.810 / 0.726 |

*Table 20.* **Per-model resistance scores of the MPA strategy under mild and intensive contamination.**

| Model | Arc-C | MMLU | TruthfulQA | GSM8K | RepliQA |
|---|---|---|---|---|---|
| | Mild / Intensive | Mild / Intensive | Mild / Intensive | Mild / Intensive | Mild / Intensive |
| Llama-3.2-3B | 0.890 / 0.876 | 0.842 / 0.810 | 0.759 / 0.665 | 0.733 / 0.735 | 0.924 / 0.900 |
| Yi-1.5-6B | 0.888 / 0.783 | 0.873 / 0.855 | 0.798 / 0.633 | 0.738 / 0.731 | 0.902 / 0.799 |
| vicuna-7b-v1.5 | 0.902 / 0.887 | 0.892 / 0.889 | 0.733 / 0.540 | 0.719 / 0.681 | 0.897 / 0.845 |
| Llama-3.1-8B | 0.887 / 0.879 | 0.865 / 0.813 | 0.825 / 0.652 | 0.686 / 0.705 | 0.732 / 0.538 |
| Falcon3-10B | 0.953 / 0.960 | 0.926 / 0.931 | 0.852 / 0.766 | 0.845 / 0.832 | 0.940 / 0.940 |
| Qwen2.5-14B | 0.944 / 0.954 | 0.924 / 0.930 | 0.884 / 0.853 | 0.810 / 0.816 | 0.875 / 0.854 |
| Phi-3-medium | 0.962 / 0.966 | 0.930 / 0.921 | 0.931 / 0.860 | 0.863 / 0.865 | 0.938 / 0.922 |
| DeepSeek-V2-Lite | 0.901 / 0.922 | 0.911 / 0.919 | 0.810 / 0.693 | 0.733 / 0.727 | 0.933 / 0.906 |
| Qwen2.5-32B | 0.963 / 0.973 | 0.952 / 0.941 | 0.924 / 0.882 | 0.754 / 0.770 | 0.892 / 0.885 |
| Yi-1.5-34B | 0.916 / 0.919 | 0.894 / 0.881 | 0.823 / 0.707 | 0.741 / 0.750 | 0.673 / 0.442 |

