# OpenReview forum: "The Emperor's New Clothes in Benchmarking? A Rigorous Examination of Mitigation Strategies for LLM Benchmark Data Contamination"
_ICML.cc/2025/Conference — ICML 2025 poster_

### Official Review · Reviewer_P7Sk · 2025-03-01

**Overall Recommendation:** 3

**Summary:**

This paper discusses a way to evaluate Benchmark Data Contamination (BDC) mitigation strategies. The authors set up two key standards, Fidelity and Contamination Resistance, as criteria of assessing reliability of each method. By following a rigorous evaluation pipeline, experiments on 10 LLMs, 5 benchmarks, and 20 BDC mitigation strategies show that no existing strategy significantly improves resistance over the vanilla "no benchmark update".

**Claims And Evidence:**

Yes, the authors set up a robust evaluation framework and provided relevant results to support their claim.

**Essential References Not Discussed:**

N/A

**Experimental Designs Or Analyses:**

Yes, the experimental design is clear. But there are some concerns regarding the soundness. Please refer to Weaknesses and Questions below.

**Methods And Evaluation Criteria:**

Yes, the authors provided an intuitive evaluation framework.

**Other Comments Or Suggestions:**

N/A - see Strengths and Weaknesses

**Other Strengths And Weaknesses:**

## Strengths
- The idea is intuitive and the experimental pipeline is robust.
- The paper reads well and is well organized

## Weaknesses
- I am not sure why high Fidelity is desired. The goal of BDC mitigation strategies is to deviate from the original benchmark, which is potentially utilized during LLM pre-training. The updated benchmark differing from the original version is not itself a problem, as long as the benchmark is evaluated on fairly. The authors did provide an example of "turning GSM8K into a history-based benchmark" in L181, but this is an excessively extreme case to justify using Fidelity as a criterion.
- If Contamination Resistance is intended to consider the advantage of using the original benchmark in fine-tuning, why not just exclude the ratio of "incorrect -> correct"? I am not sure why "correct -> incorrect" should also be excluded in the resistance ratio.
- While the overall evaluation pipeline is solid, one other concern is in the fundamental problem of the contamination detection task. The task of contamination detection itself is very difficult and is hard to verify the validity. Even if the authors used three detection methods to filter benchmarks, the following experiments building on this can be seen as unreliable.
- In summary, based on my concerns, I am not certain if the paper's claim is sound. (1) Benchmark selection is questionable, and (2) the evaluation criteria needs further justification. Thus, the claim that none of the prior work meeting reliability standards is rather misleading.

**Questions For Authors:**

N/A - see Strengths and Weaknesses

**Relation To Broader Scientific Literature:**

This work points out the weaknesses of prior BDC mitigation strategies, and calls for further research with respect to the criteria provided by the authors.

**Theoretical Claims:**

N/A

---

> ### Author Rebuttal · Authors · 2025-04-01
>
> > Q1: Why high fidelity is necessary
>
> High fidelity is necessary because a low fidelity score indicates that the updated benchmark has undergone **excessive** changes from the original benchmark, which can introduce two practical issues:
>
> (1) **Answer invalidation**: The modifications may alter the semantics of a question such that the original answer is no longer correct, requiring human annotation to ensure correctness.
>
> (2) **Difficulty or objective drift**: The updated question may no longer be appropriate for LLM evaluation. It could become too difficult, too trivial, or deviate the focus to unintended skills or knowledge domains. This requires human annotators not only to provide a new answer but also to assess whether the question remains suitable for evaluation.
>
> We provide qualitative examples of these two cases in Tables 6 and 5, respectively. Both issues contradict the goal of BDC mitigation strategies, which is to **automatically and efficiently** update benchmarks without requiring manual checks. High fidelity ensures that updated benchmarks remain aligned with the original evaluation objective and usable at scale.
>
> If our goal is to derive a *new* benchmark and then evaluate all models fairly (as you might have in mind), a high fidelity is not required. However, in this case, human annotation and evaluation for the validity of the new benchmark is often required, and it is a different setting from BDC contamination.
>
>
> > Q2: Why exclude "correct->incorrect" in contamination resistance
>
> The goal of a mitigation strategy is to enable accurate measurement of a model’s true capability, even if the model has been **contaminated by the original benchmark**. Contamination resistance is therefore designed to assess whether the **updated benchmark** can preserve this measurement.
>
> If we include "correct->incorrect" cases, we may encourage scenarios where the clean model outperforms the contaminated model on the updated benchmark, i.e., $R(M,D^S) > R(M^D,D^S)$. In practice, this would lead to **underestimating** the model’s true capacity and contradict the goal of mitigation, which is to recover reliable evaluation even after contamination. This is why **symmetric matching** is essential in our definition.
>
>
>
>
> > Q3: Benchmark selection is questionable
>
> we have made our **best effort** to filter contamination by applying three BDC detection methods from distinct categories and selecting only models regarded as uncontaminated by all three on all benchmarks. However, we acknowledge that we still cannot fully rule out the possibility of contamination.
>
> That said, we have made every effort to select benchmarks to ensure reliable conclusions. The four benchmarks we use, GSM8K, MMLU, Arc-C, and TruthfulQA, are **widely adopted** in prior BDC mitigation work [1-4]. In addition, we include the RepliQA dataset, a recently released benchmark with non-factual, fictional contexts. Its **recent release and non-factual nature** make it highly unlikely to be present in any model’s training data, making it a suitable candidate in our controlled pipeline.
>
>
> > Q4: Refining the claim to avoid misinterpretation
>
> To prevent misunderstanding, we will revise the abstract and introduction to more precisely state our claim:
>
> (1) While some *semantic-preserving* mitigation strategies (e.g., MPA and ITD) achieve significantly higher resistance scores than the vanilla case on **certain benchmarks** (e.g., MMLU, TruthfulQA, and RepliQA), no strategy consistently outperforms the vanilla case **across all benchmarks** in a **statistically significant** manner.
>
> (2) Further, although some strategies perform well on one metric, none effectively balances both fidelity and contamination resistance.
>
> -------
>
>
> [1] Clean-eval: Clean evaluation on contaminated large language models
>
> [2] Inference-time decontamination: Reusing leaked benchmarks for large language model evaluation
>
> [3] Automating dataset updates towards reliable and timely evaluation of large language models
>
> [4] Dynamic Evaluation of Large Language Models by Meta Probing Agents

---

> > ### Comment · Reviewer_P7Sk · 2025-04-02
> >
> > Thank you for the responses. I believe Q1 is an important aspect that requires detailed discussion in the paper. I suggest the authors include a section dedicated to relevant discussions in the final version.
> >
> > Other concerns are mostly addressed - I adjusted the score accordingly.
> >
> > Thank you.

---

> > > ### Author Response · Authors · 2025-04-06
> > >
> > > Thank you for taking the time to reevaluate our work and for your thoughtful feedback! We agree that Q1 raises an important point regarding the necessity of high fidelity, and we will include a comprehensive discussion in the final version to highlight its motivation and implications.
> > >
> > > If you have any further questions or suggestions, please feel free to let us know—we strive to consistently improve the quality and clarity of our paper.

---

### Official Review · Reviewer_gzkQ · 2025-03-10

**Overall Recommendation:** 4

**Summary:**

Designed a systematic and controlled pipeline to provide fine-grained and comprehensive assessment of existing benchmark data contamination mitigation strategies. They focus on a question-level study experimenting with 10 LLMs, 5 benchmarks, 20 mitagation stagetries with 2 scenarios. From this, they find that no existing strategy significantly impacts benchmark resutls.

**Claims And Evidence:**

Claim that existing BDC mitigation strategies are not sufficient, introducing fidelity and contamination resistance metrics. They provide evidence of this in section 3 and 5.

**Essential References Not Discussed:**

Might want to include the following citations in the related work: [1] "To the cutoff... and beyond? a longitudinal perspective on LLM data contamination" https://openreview.net/forum?id=m2NVG4Htxs, [2] "Bring Your Own Data! Self-Sensitivity Evaluation for Large Language Models" https://openreview.net/forum?id=k2xZYPZo34#discussion, and "Training on the test task confounds evaluation and emergence" https://openreview.net/forum?id=jOmk0uS1hl

[1] studies data contamination through the lens of time.
[2] proposes a new evaluation framework for mitigating contamination.
[3] shows that training on the test task can improve performance.

**Experimental Designs Or Analyses:**

The experimental design is solid. The only issue might be the number of LLMs evaluated.

**Methods And Evaluation Criteria:**

The paper evaluates the mitigation strategies through their two interpretable scores: Fidelity and Resistance.

**Other Comments Or Suggestions:**

For table 3, can you reconstruct this table in the appendix with an added row by weight class (including the number of models in that weight class)? I wonder if the size of the model impacts these scores. I wonder if the conclusions might be different under this view.

**Other Strengths And Weaknesses:**

Strengths:
 - Interesting finding
 - Good experimental design

Weaknesses:
  - More datasets (i.e adding code evals) and models

**Questions For Authors:**

See comments.

**Relation To Broader Scientific Literature:**

This is a good overview of existing mitigating techniques in previous literature and rigirously evaluating them.

**Theoretical Claims:**

n/a

---

> ### Author Rebuttal · Authors · 2025-04-01
>
> > W1: More datasets and models
>
> Our current study includes 10 LLMs, 5 benchmarks, 20 mitigation strategies, and 2 contamination scenarios, yielding 10×5×20×2 = 2000 evaluation results. While we believe this already provides a comprehensive analysis, we agree that including more models and benchmarks would further strengthen the reliability of our findings. We will discuss this as a limitation and explore broader coverage in future work.
>
> > Q1: Model size vs. resistance scores
>
> This is an excellent question. We appreciate the suggestion and will include an extended version of Table 3 in the appendix with model size weight information.
>
> Inspired by this question, we explore the correlation between model size and contamination resistance. We examine two perspectives: **raw resistance scores** and **resistance improvement over the vanilla baseline**. Our key finding is that **(1) larger models generally exhibit higher raw resistance scores, but (2) their relative advantages over mitigation strategies tend to diminish with scale**, as detailed below.
>
> (1) For each semantic-preserving mitigation strategy, we compute the average resistance score across all datasets and calculate its Spearman correlation with model size. **All strategies show positive correlations with model size.** This indicates even after being exposed to the original benchmark, larger models tend to preserve their evaluations on the updated benchmark, indicating higher behavioral stability.
>
> |Strategy|Corr (raw resistance) |Corr (resistance improvement)|
> |-|-|-|
> |Back-translation|0.33|-0.72|
> |Clean-Eval|0.40|-0.31|
> |Additional Incorrect Choices|0.69|-0.80|
> |Irrelevant Context|0.31|-0.70|
> |ITD|0.23|-0.31|
> |MPA|0.29|-0.14|
> |MPA-Choice + Trans-CN|0.64|-0.63|
> |MPA-Ques + Trans-CN|0.53|-0.14|
> |Choice Paraphrasing|0.68|-0.60|
> |Choices Permutation|0.68|-0.79|
> |Relevant Context|0.37|-0.36|
> |Synonym Replacement|0.33|-0.12|
> |Syntactic Modification|0.31|-0.49|
> |Translation (Chinese)|0.65|-0.14|
> |Translation (French)|0.61|0.05|
> |Typographical Perturbation|0.32|-0.39|
> |Vanilla|0.45|/|
>
> (2) However, as discussed in Section 5.1, contamination resistance should be interpreted **relative to the vanilla baseline**. To assess this, we computed the correlation between model size and the resistance improvement (i.e., the difference between the strategy's resistance and that of the vanilla baseline; averaged across all datasets). **Under this view, the correlations are mostly negative**. This indicates that the *relative effectiveness* of current mitigation strategies diminishes for larger models. It highlights the need for more robust and scalable approaches that can adapt to larger LLMs.
>
> > Additional reference
>
> Thank you for the suggestions. We will include the listed citations in the related work section for completeness.

---

> > ### Comment · Reviewer_gzkQ · 2025-04-01
> >
> > Can you show me raw values for model size vs resistance scores like Table 3? You can group Strategies by type or just select a couple strageries, but please include Paraphrasing as one of them.

---

> > > ### Author Response · Authors · 2025-04-06
> > >
> > > We provide below the raw resistance scores under both **mild** and **intensive** contamination for the vanilla case and four semantic-preserving strategies: **Synonym Replacement**, **Syntactic Modification**, **Choice Paraphrasing** (as requested), and **MPA**. Rows correspond to LLMs, and columns correspond to benchmarks.
> > >
> > > |Model|Arc-C|MMLU|TruthfulQA|GSM8K|RepliQA|
> > > |-|-|-|-|-|-|
> > > ||Mild/Intensive|Mild/Intensive|Mild/Intensive|Mild/Intensive|Mild/Intensive|
> > > |Llama-3.2-3B|0.904/0.870|0.873/0.833|0.728/0.643|0.694/0.688|0.871/0.803|
> > > |Yi-1.5-6B|0.890/0.553|0.866/0.791|0.749/0.625|0.735/0.718|0.724/0.455|
> > > |vicuna-7B|0.862/0.797|0.858/0.825|0.668/0.431|0.541/0.408|0.837/0.661|
> > > |Llama-3.1-8B|0.885/0.837|0.821/0.766|0.748/0.624|0.735/0.755|0.444/0.209|
> > > |Falcon3-10B|0.965/0.952|0.934/0.923|0.796/0.693|0.817/0.820|0.932/0.923|
> > > |Qwen2.5-14B|0.962/0.952|0.935/0.907|0.892/0.794|0.815/0.819|0.679/0.581|
> > > |Phi-3-medium|0.945/0.936|0.888/0.858|0.902/0.837|0.828/0.848|0.900/0.833|
> > > |DeepSeek-V2-Lite|0.906/0.910|0.901/0.902|0.734/0.605|0.726/0.716|0.909/0.845|
> > > |Qwen2.5-32B|0.977/0.974|0.929/0.921|0.909/0.859|0.789/0.795|0.632/0.592|
> > > |Yi-1.5-34B|0.932/0.918|0.812/0.797|0.814/0.755|0.797/0.799|0.161/0.071|
> > >
> > > **Table 1: Vanilla**
> > >
> > > |Model|Arc-C|MMLU|TruthfulQA|GSM8K|RepliQA|
> > > |-|-|-|-|-|-|
> > > ||Mild/Intensive|Mild/Intensive|Mild/Intensive|Mild/Intensive|Mild/Intensive|
> > > |Llama-3.2-3B|0.899/0.877|0.887/0.845|0.716/0.610|0.704/0.695|0.896/0.854|
> > > |Yi-1.5-6B|0.889/0.846|0.876/0.820|0.738/0.622|0.691/0.716|0.809/0.609|
> > > |vicuna-7B|0.869/0.809|0.860/0.805|0.672/0.447|0.620/0.491|0.869/0.757|
> > > |Llama-3.1-8B|0.901/0.846|0.830/0.774|0.748/0.605|0.725/0.757|0.539/0.341|
> > > |Falcon3-10B|0.957/0.956|0.944/0.929|0.786/0.683|0.813/0.810|0.952/0.939|
> > > |Qwen2.5-14B|0.957/0.949|0.936/0.908|0.890/0.788|0.815/0.816|0.762/0.712|
> > > |Phi-3-medium|0.943/0.931|0.913/0.877|0.909/0.860|0.828/0.832|0.917/0.869|
> > > |DeepSeek-V2-Lite|0.905/0.908|0.884/0.905|0.743/0.594|0.732/0.726|0.916/0.894|
> > > |Qwen2.5-32B|0.980/0.980|0.929/0.921|0.922/0.868|0.777/0.782|0.776/0.750|
> > > |Yi-1.5-34B|0.938/0.922|0.821/0.806|0.813/0.727|0.778/0.791|0.291/0.159|
> > >
> > > **Table 2: Synonym Replacement**
> > >
> > > |Model|Arc-C|MMLU|TruthfulQA|GSM8K|RepliQA|
> > > |-|-|-|-|-|-|
> > > ||Mild/Intensive|Mild/Intensive|Mild/Intensive|Mild/Intensive|Mild/Intensive|
> > > |Llama-3.2-3B|0.899/0.871|0.859/0.817|0.717/0.628|0.708/0.709|0.902/0.880|
> > > |Yi-1.5-6B|0.870/0.822|0.860/0.813|0.760/0.644|0.714/0.712|0.792/0.588|
> > > |vicuna-7B|0.886/0.817|0.871/0.838|0.657/0.459|0.640/0.543|0.873/0.749|
> > > |Llama-3.1-8B|0.891/0.849|0.841/0.775|0.756/0.641|0.710/0.747|0.506/0.320|
> > > |Falcon3-10B|0.950/0.936|0.944/0.944|0.796/0.677|0.801/0.794|0.939/0.938|
> > > |Qwen2.5-14B|0.952/0.950|0.924/0.896|0.879/0.797|0.812/0.824|0.784/0.729|
> > > |Phi-3-medium|0.937/0.928|0.894/0.869|0.889/0.835|0.817/0.829|0.906/0.863|
> > > |DeepSeek-V2-Lite|0.901/0.904|0.878/0.895|0.731/0.602|0.721/0.724|0.921/0.897|
> > > |Qwen2.5-32B|0.980/0.977|0.926/0.922|0.919/0.875|0.794/0.801|0.821/0.784|
> > > |Yi-1.5-34B|0.930/0.917|0.822/0.808|0.810/0.743|0.781/0.785|0.311/0.146|
> > >
> > > **Table 3: Syntactic Modification**
> > >
> > > |Model|Arc-C|MMLU|TruthfulQA|
> > > |-|-|-|-|
> > > ||Mild/Intensive|Mild/Intensive|Mild/Intensive|
> > > |Llama-3.2-3B|0.900/0.892|0.863/0.832|0.726/0.633|
> > > |Yi-1.5-6B|0.894/0.817|0.849/0.824|0.761/0.627|
> > > |vicuna-7B|0.882/0.832|0.860/0.839|0.685/0.449|
> > > |Llama-3.1-8B|0.893/0.848|0.835/0.776|0.770/0.654|
> > > |Falcon3-10B|0.951/0.955|0.930/0.928|0.797/0.698|
> > > |Qwen2.5-14B|0.956/0.955|0.937/0.907|0.880/0.800|
> > > |Phi-3-medium|0.933/0.941|0.891/0.875|0.908/0.857|
> > > |DeepSeek-V2-Lite|0.896/0.904|0.894/0.902|0.737/0.608|
> > > |Qwen2.5-32B|0.978/0.980|0.928/0.927|0.897/0.870|
> > > |Yi-1.5-34B|0.928/0.914|0.850/0.817|0.810/0.726|
> > >
> > > **Table 4: Choice Paraphrasing (only 3 multiple-choice benchmarks available)**
> > >
> > > |Model|Arc-C|MMLU|TruthfulQA|GSM8K|RepliQA|
> > > |-|-|-|-|-|-|
> > > ||Mild/Intensive|Mild/Intensive|Mild/Intensive|Mild/Intensive|Mild/Intensive|
> > > |Llama-3.2-3B|0.890/0.876|0.842/0.810|0.759/0.665|0.733/0.735|0.924/0.900|
> > > |Yi-1.5-6B|0.888/0.783|0.873/0.855|0.798/0.633|0.738/0.731|0.902/0.799|
> > > |vicuna-7B|0.902/0.887|0.892/0.889|0.733/0.540|0.719/0.681|0.897/0.845|
> > > |Llama-3.1-8B|0.887/0.879|0.865/0.813|0.825/0.652|0.686/0.705|0.732/0.538|
> > > |Falcon3-10B|0.953/0.960|0.926/0.931|0.852/0.766|0.845/0.832|0.940/0.940|
> > > |Qwen2.5-14B|0.944/0.954|0.924/0.930|0.884/0.853|0.810/0.816|0.875/0.854|
> > > |Phi-3-medium|0.962/0.966|0.930/0.921|0.931/0.860|0.863/0.865|0.938/0.922|
> > > |DeepSeek-V2-Lite|0.901/0.922|0.911/0.919|0.810/0.693|0.733/0.727|0.933/0.906|
> > > |Qwen2.5-32B|0.963/0.973|0.952/0.941|0.924/0.882|0.754/0.770|0.892/0.885|
> > > |Yi-1.5-34B|0.916/0.919|0.894/0.881|0.823/0.707|0.741/0.750|0.673/0.442|
> > >
> > > **Table 5: MPA**
> > >
> > >
> > > **Due to space limitations, we only present a subset of strategies here. If there are additional strategies you are interested in, please feel free to let us know—we would be glad to provide the corresponding results or include them in the final appendix.**

---

### Official Review · Reviewer_Eeab · 2025-03-13

**Overall Recommendation:** 2

**Summary:**

This paper introduces a systematic pipeline and proposes two metrics—fidelity and contamination resistance—to provide a fine-grained and comprehensive assessment of existing benchmark data contamination (BDC) mitigation strategies. The authors evaluated 20 different BDC mitigation approaches across 10 LLMs, 5 benchmarks, and 2 contamination scenarios, and found that none of the existing mitigation strategies consistently improved contamination resistance across all benchmarks while maintaining fidelity to the original tests

**Claims And Evidence:**

There are some contradictory claims, for instance:

a. The authors mention that no existing BDC mitigation strategy is effective. However, the results show that some strategies (e.g., MPA, ITD, and Analysis Extension) significantly outperform the vanilla (no mitigation) approach

b. The paper assumes that existing benchmarks (e.g., MMLU, GSM8K) are high-quality but there are already updated versions of these benchmarks due to issues like wrong labels, data contamination, etc. [1, 2].

1. Zhang, Hugh, et al. "A careful examination of large language model performance on grade school arithmetic." Advances in Neural Information Processing Systems 37 (2024): 46819-46836.

2. Wang, Yubo, et al. "Mmlu-pro: A more robust and challenging multi-task language understanding benchmark." The Thirty-eight Conference on Neural Information Processing Systems Datasets and Benchmarks Track. 2024.

**Essential References Not Discussed:**

N/A.

**Experimental Designs Or Analyses:**

Contamination scenarios are quite synthetic, making the generalizability of the results questionable.

**Methods And Evaluation Criteria:**

1. While the proposed methods and evaluation criteria are not something groundbreaking (i.e., instead of applying approaches like paraphrasing, fine-tuning, etc.), they still make sense.
2. The authors also fail to justify the novelty of their proposed metrics.

**Other Comments Or Suggestions:**

N/A

**Other Strengths And Weaknesses:**

Strength:

1. Studied a very important topic.

Weaknesses:

1. Wrong claims, lacking substantial contributions,

**Questions For Authors:**

a. Why did you mention that no existing BDC mitigation strategy is effective even though your experiment shows contradictory results?

b. Why did you state that existing benchmarks (e.g., MMLU, GSM8K) are high-quality but there are already updated versions of these benchmarks due to issues like wrong labels, data contamination, etc.

c. What is the technical novelty of the proposed metrics?

**Relation To Broader Scientific Literature:**

1. Contamination scenarios are quite synthetic, making the generalizability of the results questionable.

**Theoretical Claims:**

1. No issues.

---

> ### Author Rebuttal · Authors · 2025-04-01
>
> > W1 & Q1: Misunderstanding of contradictory results
>
> We clarify that our claim is not contradictory: while some **semantic-preserving** mitigation strategies (e.g., MPA and ITD) achieve significantly higher resistance scores than the vanilla case on **certain benchmarks** (e.g., MMLU, TruthfulQA, and RepliQA), no strategy consistently outperforms the vanilla case **across all benchmarks** in a **statistically significant** manner (Table 3). Our conclusion excludes semantic-altering strategies (e.g., Analysis Extension), which are only applicable to Arc-C and MMLU and thus insufficient to support a general claim.
>
> To prevent misunderstanding, we will revise the abstract and introduction to make the claim more accurate.
>
> > Q2: Justifying the use of existing benchmarks in BDC mitigation evaluation
>
> We emphasize that BDC mitigation is an established line of research [1–4], which builds on the assumption that widely used benchmarks such as MMLU and GSM8K are high-quality and representative of real-world question distributions. These benchmarks are commonly adopted in prior work proposing mitigation strategies to address contamination. **Building upon their assumptions**, our paper rigorously examines the effectiveness of such strategies.
>
> High quality does not imply perfection. Rather, it suggests that these benchmarks have broad coverage aligned with the intended evaluation objectives, making them worth preserving through mitigation rather than replacement. While we acknowledge that benchmarks like MMLU and GSM8K may contain incorrect labels and suffer from data contamination, even **their revised versions** (e.g., MMLU-Pro, GSM1K) **remain vulnerable to contamination**. This further motivates the need for robust contamination mitigation strategies and careful evaluation of their effectiveness.
>
> We thank the reviewer for highlighting this point and the relevant references, and will cite them and include a discussion in the revision.
>
> > Q3: Novelty of the proposed metrics
>
> We identify clear limitations in existing practices for assessing BDC mitigation strategies: (1) Accuracy drop **ignores clean accuracy**, making it unclear how much drop reflects effective mitigation; (2) Accuracy matching focuses on aggregate accuracy but **overlooks question-level mismatches**. For example, a strategy with high accuracy matching may still alter the original benchmark’s evaluation objective (Fig 2(b)).
>
> Motivated by these issues, we propose two metrics, fidelity and contamination resistance, that explicitly capture **two** types of **question-level alignment** using normalized Hamming distance. To our knowledge, this is the first work to examine BDC mitigation effectiveness along these two orthogonal dimensions, covering more desirable aspects of mitigation strategies and enabling finer-grained analysis than prior approaches.
>
>
> > Experimental Designs Or Analysis: On the realism of contamination scenarios
>
> We included two contamination settings in our paper that reflect **common and established** practices in prior BDC mitigation work: (a) Intensive Contamination [1,3]: fine-tuning the LLM with only benchmark data. (b) Mild Contamination [4]: fine-tuning on benchmark data mixed with 20K instruction-following samples from OpenOrca.
>
> Motivated by your concern, we **add experiments** on two more scenarios: (c) Partial Contamination: only **half of the benchmark** is included in fine-tuning, mixed with 20K OpenOrca samples, while evaluation is done on the **entire benchmark**. This reflects situations where only a portion of the evaluation data is seen during training. (d) Indirect Contamination: fine-tuning and evaluation use different splits of the same benchmark, again mixed with 20K OpenOrca samples. This setting captures contamination via exposure to data from the same distribution during training, without direct sample overlap with evaluation data.
>
> We experiment with 8 LLMs and 2 datasets, evaluating all 16 semantic-preserving strategies. We report only those strategies (with resistance scores) that achieve **statistically significantly higher resistance than Vanilla**; full results will be included in the revised appendix.
>
> (1) Partial contamination:
> - Arc-C: No strategy shows significant improvement over Vanilla;
> - TruthfulQA: Back Translation (0.807), ITD (0.824), MPA (0.833), and MPA-Ques + Trans-CN (0.813) outperform Vanilla (0.795).
>
> (2) Indirect contamination:
> - Arc-C: No strategy shows significant improvement over Vanilla;
> - TruthfulQA: ITD (0.821), MPA (0.839), and MPA-Ques + Trans-CN (0.815) outperform Vanilla (0.791).
>
> These empirical results echo with our main claim (refer to W1 & Q1).
>
> ---
>
>
> [1] Clean-eval: Clean evaluation on contaminated large language models
>
> [2] Dynamic evaluation of large language models by meta probing agents
>
> [3] Automating dataset updates towards reliable and timely evaluation of large language models
>
> [4] ConStat: Performance-Based Contamination Detection in Large Language Models

---

### Official Review · Reviewer_2iU4 · 2025-03-14

**Overall Recommendation:** 4

**Summary:**

This paper investigates mitigation strategies for benchmark data contamination (BDC) in LLM evaluation. The authors argue that current approaches for assessing BDC mitigation strategies, which focus on aggregate accuracy metrics, have significant limitations. To address this, they propose two metrics---fidelity and contamination resistance---that enable question-level evaluation. Experiments with 10 LLMs, 5 benchmarks, and 20 mitigations strategies show that no existing strategy consistently outperforms the vanilla approach (i.e. no dataset update) across all benchmarks, and none balances both fidelity and contamination resistance.

**Claims And Evidence:**

The claims are generally well-supported by evidence. The authors claim that previous BDC mitigation assessment methods are insufficient is argued through examples in Figure 2, and show why question-level matching is more informative than aggregate accuracy.

The central claim that no existing strategy significantly improves resistance over the vanilla case across all benchmarks is supported by the results in Tables 3 and 4, with statistical significance testing. The data shows that while some strategies perform well on specific benchmarks, none consistently outperforms across all datasets.

The claim regarding the trade-off between fidelity and resistance is also shown  in Figure 4, where strategies are visibly clustered in either high-fidelity/low-resistance or low-fidelity/high-resistance regions, with none achieving both high fidelity and high resistance.

**Essential References Not Discussed:**

Recent work on red-teaming LLMs and adversarial robustness provide some insights on developing perturbation techniques that maintain semantic equivalence while bypassing pattern recognition. These approaches directly relate to the semantic-preserving strategies examined in the paper.

**Experimental Designs Or Analyses:**

The experimental design is well-constructed with appropriate controls like three BDC detection methods to ensure uncontaminated baseline models, two different contamination scenarios, validation of contamination effectiveness, and monitoring model perplexity on held out data. One potential limitation is the focus on a specific finetuning approach for contamination.

**Methods And Evaluation Criteria:**

The methodology is appropriate and includes uncontaminated LLM-benchmark pair selection using three BDC detection methods, application of 20 mitigation strategies, controlled contamination under two scenarios (mild and intensive), and evaluation using the proposed metrics.

The selection of benchmarks (Arc-Challenge, MMLU, TruthfulQA, GSM8K, and RepliQA) and models (10 LLMs ranging from 3B to 34B parameters) provides good coverage of different evaluation contexts.

**Other Comments Or Suggestions:**

It would be interesting to see some discussion probablistic evaluation metrics that account for uncertainty in model responses are alternatives to the binary evaluation vectors used in the paper's metrics and could potentially offer a more nuanced assessment of contamination effects.

**Other Strengths And Weaknesses:**

Other weaknesses:
- The benchmarks and experiments used are primarily multiple-choice and more straightforward open-ended questions, so it is unclear whether the findings generalizes to more complex evaluation tasks.
- While the experiments cover 10 LLMs, they are all relatively small (3B to 34B parameters) compared to the SOTA models, which raises questions about if and how the results generalizes to larger models.

**Questions For Authors:**

1. Have you noticed any patterns in the types of questions that benefit most from specific mitigation strategies?
2. Any ideas for mitigation strategies that would more effectively balance fidelity and resistance?
3. Have you explored/observed any correlation between the model size and their susceptibility to contamination and the responsiveness to mitigation strategies?

**Relation To Broader Scientific Literature:**

The paper is well-situated within the broader literature on LLM evaluation and benchmark contamination. The authors acknowledge two primary approaches to addressing BDC: creating new benchmarks and updating existing ones, focusing on the latter as a more cost-effective approach. They build upon previous mitigation strategies like Clean-Eval, ITD, and MPA while addressing limitations in their evaluation methodologies. The discussion of BDC detection methods is comprehensive, categorizing them into token probability-based, generation-based, and order-based approaches.

**Theoretical Claims:**

The paper focuses on empirical evaluation and makes limited formal theoretical claims. The definitions of fidelity and contamination resistance in Section 3 are mathematically sound, as is the extension to continuous evaluation scores in Appendix A.1.

---

> ### Author Rebuttal · Authors · 2025-04-01
>
> > W1: More complex evaluation tasks
>
> Thank you for the insightful suggestion. As the first work to rigorously assess BDC mitigation strategies for LLMs, we focus on commonly used evaluation tasks as adopted in prior BDC mitigation studies [1-3]. We agree that extending the analysis to more complex evaluation tasks is an important direction. We will discuss this limitation in the revised paper and explore it in future work.
>
> > W2: Larger LLMs
>
> We agree that including larger models would further strengthen the reliability of our findings. However, our setup adopts **full fine-tuning** because it more faithfully approximates real-world contamination scenarios, where models are exposed to benchmark data during pre-training or continued training, compared to parameter-efficient methods like LoRA. However, full fine-tuning can be computationally expensive for larger LLMs.
>
> Notably, most prior works on BDC mitigation [1-2] only consider LLMs with up to 13B parameters. In comparison, our study includes models up to 34B, and we have made every effort to scale as far as our resources allow. We plan to include larger models in future work as resources permit.
>
> > S1: Discussion of probabilistic evaluation metrics
>
> Thank you for the valuable perspective. We will include a discussion of this point in the revised paper and consider it as part of future work.
>
> > Q1: Any patterns in the types of questions that benefit most?
>
> For multiple-choice benchmarks, we find that ITD, MPA, and choice permutation achieve high contamination resistance. In contrast, for open-ended benchmarks, we do not observe any strategy that consistently and statistically significantly outperforms the vanilla baseline in terms of contamination resistance. This may be due to the greater variability and flexibility of open-ended responses, which makes stable mitigation more difficult.
>
>
> > Q2: Toward strategies that better balance fidelity and resistance
>
> One potential direction we are exploring in separate work involves training two reward models, one for fidelity and one for resistance, and using them to jointly guide the LLM in conditionally updating benchmarks that score highly on both axes. This remains an open question without a definitive solution, and we believe learning-based approaches offer a potential path forward.
>
> > Q3: Correlation between model size and (1) susceptibility to contamination and (2) responsiveness to mitigation
>
> This is an excellent question. (1) Inspired by it, we compute the Spearman correlation between model size and accuracy inflation (averaged across 5 datasets) under mild and intensive contamination. We observe a negative correlation, suggesting that larger models exhibit less accuracy inflation under contamination. One possible explanation is that their stronger generalization capabilities make them less dependent on memorized benchmark content.
>
> |Contamination|Corr (**averaged** across 5 datasets)|
> |-|-|
> |Mild|-0.018|
> |Intensive|-0.401|
>
> (2) We further explore the correlation between model size and contamination resistance. Specifically, for each semantic-preserving mitigation strategy, we compute the average resistance score across all datasets and calculate its Spearman correlation with model size. **All strategies show positive correlations with model size.** This indicates even after being exposed to the original benchmark, larger models tend to preserve their evaluations on the updated benchmark, indicating higher behavioral stability.
>
> |Strategy|Corr (raw resistance) |Corr (resistance improvement)|
> |-|-|-|
> |Back-translation|0.33|-0.72|
> |Clean-Eval|0.40|-0.31|
> |Additional Incorrect Choices|0.69|-0.80|
> |Irrelevant Context|0.31|-0.70|
> |ITD|0.23|-0.31|
> |MPA|0.29|-0.14|
> |MPA-Choice + Trans-CN|0.64|-0.63|
> |MPA-Ques + Trans-CN|0.53|-0.14|
> |Choice Paraphrasing|0.68|-0.60|
> |Choices Permutation|0.68|-0.79|
> |Relevant Context|0.37|-0.36|
> |Synonym Replacement|0.33|-0.12|
> |Syntactic Modification|0.31|-0.49|
> |Translation (Chinese)|0.65|-0.14|
> |Translation (French)|0.61|0.05|
> |Typographical Perturbation|0.32|-0.39|
> |Vanilla|0.45|/|
>
> However, as discussed in Section 5.1, contamination resistance should be interpreted **relative to the vanilla baseline**. To assess this, we computed the correlation between model size and the resistance improvement (i.e., the difference between the strategy's resistance and that of the vanilla baseline; averaged across all datasets). Under this view, **the correlations are mostly negative**. This indicates that the *relative effectiveness* of current mitigation strategies diminishes for larger models. It highlights the need for more robust and scalable approaches that can adapt to larger LLMs.
>
> ---
>
> [1] Clean-eval: Clean evaluation on contaminated large language models
>
> [2] Automating dataset updates towards reliable and timely evaluation of large language models
>
> [3] Dynamic Evaluation of Large Language Models by Meta Probing Agents

---

### Decision · Program_Chairs · 2025-05-01

**Decision:**

Accept (poster)

**Comment:**

This paper studies an important problem in LLMs, i.e., the assessment of existing BDC mitigation strategies. To do the assessment, it designs a pipeline with two metrics—fidelity and contamination resistance. Extensive experiments have been conducted. These findings highlight the need for designing more effective BDC mitigation strategies.